# Stereotyped transcriptomic transformation of somatosensory neurons in response to injury

**Minh Q Nguyen[1], Claire E Le Pichon[2], Nicholas Ryba[1]\***

[1]National Institute of Dental and Craniofacial Research, Bethesda, United States; [2]National Institute of Child Health and Human Development, Bethesda, United States

**Abstract** In mice, spared nerve injury replicates symptoms of human neuropathic pain and induces upregulation of many genes in somatosensory neurons. Here we used single cell transcriptomics to probe the effects of partial infraorbital transection of the trigeminal nerve at the cellular level. Uninjured neurons were unaffected by transection of major nerve branches, segregating into many different classes. In marked contrast, axotomy rapidly transformed damaged neurons into just two new and closely-related classes where almost all original identity was lost. Remarkably, sensory neurons also adopted this transcriptomic state following various minor peripheral injuries. By genetically marking injured neurons, we showed that the injury-induced transformation was reversible, with damaged cells slowly reacquiring normal gene expression profiles. Thus, our data expose transcriptomic plasticity, previously thought of as a driver of chronic pain, as a programed response to many types of injury and a potential mechanism for regulating sensation during wound healing.

DOI: https://doi.org/10.7554/eLife.49679.001

**\*For correspondence:**
nick.ryba@nih.gov

**Competing interests:** The authors declare that no competing interests exist.

## Introduction

The somatosensory system is responsible for providing vertebrates with important information about thermal, chemical and mechanical cues by eliciting a very wide range of distinct sensations that allow animals to respond appropriately (*Abraira and Ginty, 2013*; *Basbaum et al., 2009*; *Le Pichon and Chesler, 2014*). For example, noxious stimuli activate nociceptors triggering spatially localized pain that serves to inform both immediate responses and future avoidance of risky behavior (*Basbaum et al., 2009*; *Julius, 2013*). However, maladaptive transformation of somatosensation can lead to chronic pain. For example, in human subjects, various types of nerve damage result in neuropathic pain, which rather than providing essential information about the external environment greatly impacts quality of life (*Basbaum et al., 2009*; *Colloca et al., 2017*; *Costigan et al., 2009*). This type of pain is often modeled in animals by spared nerve injury (SNI) where major branches of a peripheral nerve are crushed or cut (*Cobos et al., 2018*; *Costigan et al., 2009*; *Guan et al., 2016*). Many studies have demonstrated that such lesion of the sciatic nerve and partial infraorbital trigeminal nerve transection (IOT) result in the development of mechanical allodynia (where gentle touch elicits pain behavior) and cold sensitivity (*Basbaum et al., 2009*; *Choi et al., 2016*; *Cobos et al., 2018*; *Guan et al., 2016*; *Hardt et al., 2019*; *Ito et al., 2013*; *Xu et al., 2008*). Just as in human neuropathic pain symptoms typically develop slowly, over a period of days, and then persist long-term.

One well-characterized effect of SNI is the upregulation of genes in response to injury (*Cobos et al., 2018*; *Guan et al., 2016*; *Shin et al., 2019*; *Wlaschin et al., 2018*). These genes encode neuropeptides such as galanin (*Gal*), neuropeptide Y (*Npy*) and neurotensin (*Nts*),

transcription control factors including activating transcription factor 3 (*Atf3*), SRY-Box11 (*Sox11*) and cytokines like colony stimulating factor 1 (*Csf1*) that are all expressed in injured neurons. It is generally thought that these transcriptional changes are important for triggering neuropathic pain with recent work showing that upstream inhibition of this gene expression program attenuated pain (*Wlaschin et al., 2018*). More specifically, upregulation of *Csf1* expression after nerve injury has been genetically linked to development of chronic pain (*Guan et al., 2016*). However, the significance for most of the SNI induced gene expression changes remains to be determined.

Somatosensory neurons are diverse with wide-ranging conduction velocities, cell soma diameters, expression profiles of receptors, ion-channels and neuropeptides; they have select peripheral and central targets and also exhibit varied functional response profiles (*Julius, 2013*; *Le Pichon and Chesler, 2014*; *Nguyen et al., 2017*; *Zimmerman et al., 2014*). Does the identity of a damaged neuron affect its transcriptional response to SNI? And are uninjured neurons affected at a transcriptional level? We reasoned that answering these questions would help address the etiology of neuropathic pain and used single nucleus (sn)-based RNA sequencing (*Lacar et al., 2016*) to examine the transcriptomic response to IOT at the level of the individual neurons. Our results revealed a dramatic change of gene expression in damaged neurons that transformed diverse classes of trigeminal neurons into a uniform and distinct injury-related state. Quite unexpectedly, we discovered that various types of minor peripheral injury induce very similar transcriptomic changes to IOT. Unlike SNI, injuries of this type do not trigger chronic pain. Thus, although some of the genes that are overexpressed following nerve damage may trigger neuropathic pain in extreme cases (*Guan et al., 2016*; *Wlaschin et al., 2018*), the physiological role for this transcriptional transformation of damaged neurons appears to be much more general and provides a mechanism for controlling somatosensory and painful input from sites of injury.

## Results and discussion

### sn-RNA sequencing provides an unbiased classification of trigeminal neurons

Single cell sequencing of neurons from the dorsal root and trigeminal ganglia has identified about a dozen cell-types and great similarity between the ganglia (*Chiu et al., 2014*; *Gatto et al., 2019*; *Li et al., 2016*; *Nguyen et al., 2017*; *Usoskin et al., 2015*). However, large diameter neurons expressing genes such as the mechanosensory channel *Piezo2*, neurofilament heavy polypeptide (*Nefh*), and protein S100β (*S100b*) were poorly represented in Dropseq sequencing of trigeminal neurons (*Nguyen et al., 2017*) probably because of selective loss of larger neurons during the dissociation and capture of single cells. We reasoned that transcriptomic analysis of nuclei (*Lacar et al., 2016*), which does not require cell purification (*Figure 1a*), might provide a less biased description of the neural complement of the trigeminal ganglion and used a targeted nuclear sequencing approach (*Figure 1a*) to selectively analyze sn-transcriptomes from more than 7500 neurons (*Figure 1b*). Major divisions between trigeminal neural-types that had been characterized by single cell sequencing (*Nguyen et al., 2017*) were segregated in the sn-analysis as distinct clusters expressing diagnostic markers (*Figure 1c*). Data from the nuclei much more closely matched the neural composition of the trigeminal ganglion (*Nguyen et al., 2017*) and were dominated by *S100b*-expressing large diameter neurons that now could be divided into several new sub-classes (*Figure 1c*). For simplicity, we have designated these new sub-classes of large diameter neurons by their relationship to the classes identified in the earlier analysis.

Since cells do not need to be dissociated and isolated for sn-transcriptome analysis there is little processing time during which gene-expression can change because of neuronal damage or stress. The benefit of using nuclear rather than cellular RNA sequencing and reduced handling was demonstrated by the accurate representation of immediate early genes like the proto-oncogene c-Fos (*Fos*) and early growth response protein 1 (*Egr1*) in the sn-data. Both genes were prominent in the earlier single cell analysis (*Figure 1—figure supplement 1a*; *Nguyen et al., 2017*). By contrast, in situ hybridization (ISH) did not detect at high level or a significant number of cells expressing these genes (*Figure 1—figure supplement 1b*), which instead matched data from sn-sequencing. Importantly, genes that are thought to be markers of neural damage such as *Atf3* and *Gal* also corresponded to in situ hybridization results in the nuclei-based dataset but were much more frequently

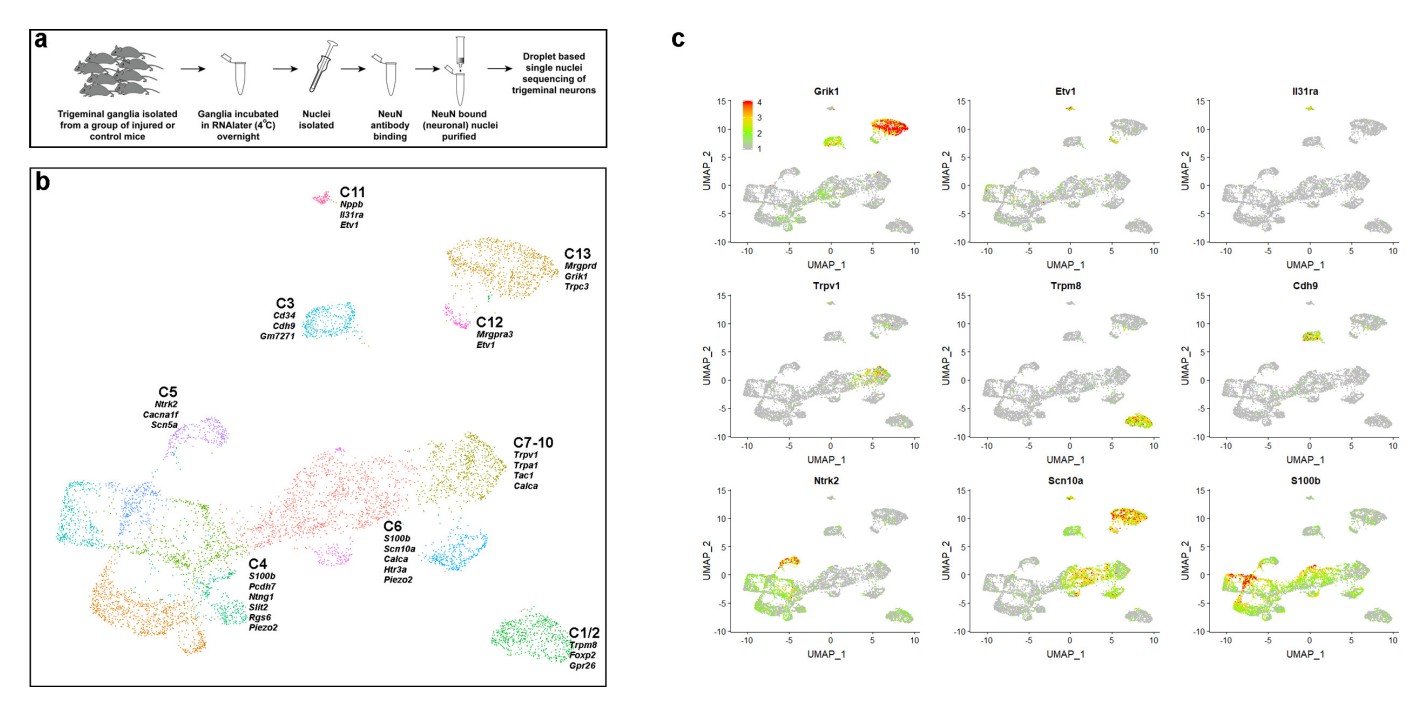

**Figure 1.** sn-RNA sequence analysis of trigeminal neurons from uninjured control mice. (**a**) Schematic representation of strategy used for selective sn-sequencing of trigeminal neurons. (**b**) UMap representation of cluster analysis for sn-data from 7546 trigeminal neurons; each cluster is assigned a distinct color. The designations (C1–C13) are based on earlier single cell data (**Nguyen et al., 2017**) and do not distinguish between the new classes of cells identified here (see **Supplementary file 4** for detailed description of trigeminal neural classes). Several prominent genes that help distinguish the classes are listed. (**c**) Expression profiles of select genes in the UMap representation with relative cellular expression colored as indicated by the scale-bar. The expression profiles highlight the segregation of gene expression between clusters.

DOI: https://doi.org/10.7554/eLife.49679.002

The following figure supplement is available for figure 1:

**Figure supplement 1.** Isolation of trigeminal neurons modifies cellular expression of genes.

DOI: https://doi.org/10.7554/eLife.49679.003

detected in sequencing of isolated cells (**Figure 1—figure supplement 1**). Thus sn-sequencing provides a valuable platform for studying changes in gene expression induced in response to cellular stress and damage as, for example, following SNI.

## Rapid induction of a shared injured-neuron transcriptional state after peripheral axotomy

SNI causes changes in gene expression that are thought to play an important role in injury-related tactile allodynia and neuropathic pain (**Basbaum et al., 2009**; **Costigan et al., 2009**; **Guan et al., 2016**; **Wlaschin et al., 2018**). To study these changes at the level of individual neurons after nerve damage, mice were subjected to IOT and the ipsilateral trigeminal ganglion was isolated for sn-transcriptomics (**Lacar et al., 2016**). We began by examining transcriptional changes two days after nerve transection (**Figure 2**). In most SNI models, this time-point precedes maximal injury-related changes in mechanical allodynia (**Cobos et al., 2018**; **Hardt et al., 2019**; **Wlaschin et al., 2018**) but already induces expression of injury-related markers. Since IOT cuts just a subset of trigeminal fibers we expected that only a fraction of sn-transcriptomes would express injury induced genes. Indeed, most trigeminal neurons displayed a normal array of diverse gene expression profiles, clustering into classes (**Figure 2a**) corresponding to those seen in the uninjured control (**Figure 1**). By contrast, very few damaged neurons expressing *Atf3* were associated with these standard trigeminal neural classes and, instead, clustered together in two new related groups, I1 and I2 that were well separated from uninjured neurons in the UMap projection (**Figure 2a,b**). In combination, I1 and I2 accounted for 635 of 4611 sequenced neurons (approx. 13.8%). Transcriptome analysis revealed that the injured

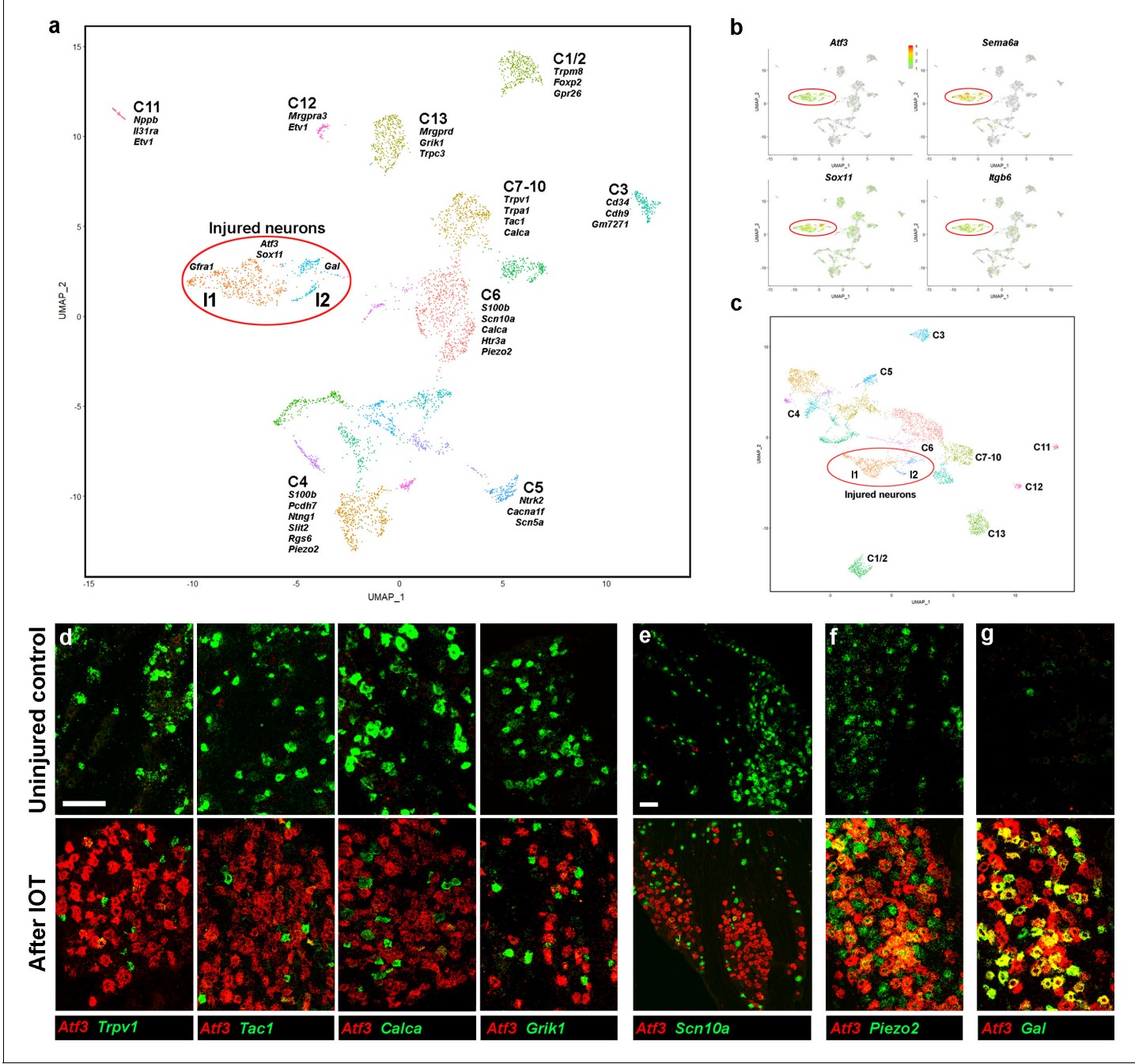

**Figure 2.** Analysis of trigeminal neurons two days after IOT. (**a**) UMap representation of sn-cluster analysis from 4611 trigeminal neurons isolated 2 days after IOT. Each cluster is assigned a distinct color and new injury-related clusters I1 and I2 are circled. The designations of uninjured neurons (C1–C13) are based on earlier single cell data (*Nguyen et al., 2017*) and do not distinguish between the new classes of cells identified here (see *Supplementary file 4* for detailed description of trigeminal neural classes). (**b**) Expression profiles of select injury-related genes (relative cellular expression colored as indicated by the scale-bar) highlight strong upregulation of these genes in the I1 and I2 clusters (circled). (**c**) Re-clustered data: excluding genes that were prominently up- or down-regulated in injured neurons from analysis. I1 and I2 class cells (circled) still segregate from uninjured cell classes. (**d–g**) Representative images of double label ISH for *Atf3* (red) and select genes (green) in trigeminal ganglia from control mice (upper panels) and animals 2 days after IOT (lower panels), illustrate (**d, e**) down-regulation of expression of several key genes (green) in injured neurons labeled by *Atf3* (red). (**f**) There was less effect on expression of *Piezo2* (green) in *Atf3*-positive cells (red); (**g**) *Gal* (green) was detected in a subset of injured neurons expressing *Atf3* (red). (**e**) Lower magnification image showing that *Scn10a* (green) is almost totally excluded from regions of the ganglion where many neurons express *Atf3* (red); scale bars = 100 μm for both magnifications; see *Supplementary file 1* for quantitation and statistical analysis of ISH data.

DOI: https://doi.org/10.7554/eLife.49679.005

*Figure 2 continued on next page*

*Figure 2 continued*

The following figure supplements are available for figure 2:

**Figure supplement 1.** Characterization of trigeminal neurons two days after spared nerve injury.

DOI: https://doi.org/10.7554/eLife.49679.006

**Figure supplement 2.** Differences between I1 and I2 classes of injured neurons and relationship to uninjured neurons Gene expression in the sn-analysis shown in *Figure 2* (relative expression indicated by scale-bar).

DOI: https://doi.org/10.7554/eLife.49679.007

**Figure supplement 3.** GO-analysis of genes that are differentially expressed in I1 and I2.

DOI: https://doi.org/10.7554/eLife.49679.008

neurons over-expressed a series of genes (*Figure 2b* and *Figure 2—figure supplement 1a*) including many that have been previously linked to nerve damage (*Table 1a*). We also confirmed this at a cellular level using ISH to co-localize several different injury-related markers (*Figure 2—figure supplement 1b*). However, since injury-related changes reflect the upregulation of a common transcriptional program (*Wlaschin et al., 2018*), it is possible that expression of this shared set of genes masks underlying differences between the injured neurons and hides residual similarity to their original identities. To investigate if this was the case, we re-clustered the sn-data excluding genes that were identified as up- or down-regulated in the *Atf3*-positive neural class from the clustering (see Materials and methods for detail). Remarkably, even without these injury-related transcripts playing a role, *Atf3*-expressing neurons still grouped together as two related classes and remained segregated from the uninjured trigeminal neural classes (*Figure 2c*, *Figure 2—figure supplement 1c*).

Given that injured neurons segregate from standard trigeminal neural classes even when the most diagnostic markers are ignored, we reasoned that many transcripts that normally define somatosensory neural classes must be dramatically down-regulated in the damaged cells. The expression heatmap of trigeminal markers (*Figure 2—figure supplement 1a*) confirmed this was generally the case. We next used ISH to independently examine down-regulation of key genes at a cellular level after nerve injury. The chosen genes, *Trpv1*, protachykinin 1 (*Tac1*), calcitonin gene-related peptide 1 (*Calca*), kainate one receptor (*Grik1*) and *Scn10a* are normally expressed at high levels in significant numbers of trigeminal neurons (*Figure 2d,e* upper panels). These genes were far less often present in *Atf3*-positive cells (*Figure 2d,e*, lower panels, see *Supplementary file 1* for quantitation and statistical analysis). In addition, the few *Atf3*-positive cells expressing these transcripts typically exhibited reduced signal intensity when compared to the surrounding uninjured neurons (*Figure 2d,e*). Notably, for *Scn10a*, which is expressed in nearly half of the trigeminal neurons (*Figure 1b*, *Figure 2e*), this led to distinctive holes in its expression pattern across the regions of the ganglion where *Atf3*-positive neurons were strongly clustered (*Figure 2e*). Thus, in combination our data demonstrate that within 2 days of injury there is upregulation of a shared set of injury markers and strong downregulation of many genes that are normally prominent markers of the different trigeminal neural classes.

Somatosensory neurons exhibit a diverse array of functions, conduction velocities, myelination states and central projections (*Basbaum et al., 2009*; *Le Pichon and Chesler, 2014*; *Zimmerman et al., 2014*), therefore the rapid collapse of all transcriptomic classes to a much less complex injured state was by far the most remarkable feature of our data. However, cluster analysis indicated that some differences between injured neurons divided them into two distinct but related classes: I1 and I2 (*Figure 2a*). Several genes that broadly distinguish groups of uninjured trigeminal neurons were also differentially represented in I1 and I2 (see *Figure 2—figure supplement 2* for examples and their expression profiles in the injured and uninjured neural classes). I1 neurons typically expressed markers including protocadherin 7 (*Pcdh7*), *Piezo2*, netrin G1 (*Ntng1*), microtubule associated protein 2 (*Map2*) and FAT atypical cadherin 2 (*Fat3*) that normally are prominent in larger diameter neurons involved in more discriminative types of sensation. ISH confirmed that a large subset of injured neurons still expressed *Piezo2* (*Figure 2f*). I2 neurons had an expression profile more related to nociceptors and instead expressed genes like tee-shirt homology domain 2 (*Tshz2*), regulator of G protein signaling 4 (*Rgs4*) and calcium voltage gated channel subunit alpha 1 c (*Cacna1c*) that are normally markers of smaller diameter neurons (*Figure 2—figure supplement 2*). Finally, by comparing the expression profiles of I1 and I2 neurons, we identified a few genes that were differentially upregulated after injury. For example, GDNF-family receptor 1 (*Gfra1*) was primarily expressed

**Table 1.** Genes up- and down-regulated in injured trigeminal neurons.
Genes that are affected by IOT were identified by comparing expression in the two injured neuron classes relative to other classes. (a) Selected up-regulated genes, their proposed functional role and previous assignment as injury-related transcripts (*Cobos et al., 2018*; *Guan et al., 2016*; *Shin et al., 2019*; *Wlaschin et al., 2018*) and/or as genes involved in neural regeneration (*Mahar and Cavalli, 2018*). (b) Selected down-regulated genes, their functional classification and previous assignment as injury-related transcripts (*Shin et al., 2019*). See *Supplementary file 2*; *Supplementary file 3*; *Supplementary file 4* for more information about genes expression changes in injured neurons including magnitude of changes and probability that expression is altered and *Figure 2—figure supplement 3* for GO-analysis.

**Table 1a**

| Gene | Reported functional roles | Reported previously[1,2] |
|---|---|---|
| Atf3 | Transcriptional regulation: (up and downregulation) ATF/CREB family of transcription factors | a, b, c, D |
| Sema6a | Semaphorin: receptor for Plxna2 role in cell-cell signaling and appropriate neural projection | a, b, c |
| Sox11 | Transcription factor: regulates survival and axonal growth in embryonic sensory neurons | a, b, D |
| Gal | Neuropeptide: modulation and inhibition of action potentials | a, b, c |
| Npy | Neuropeptide: neuropeptide with multiple roles including reducing pain perception | a, b, c |
| Nts | Neuropeptide: multiple roles in neurotransmission/modulation | a, b, c |
| Mmp16 | Metallopeptidase: extracellular matrix degradation | a, b, c |
| Itgb6 | Integrin subunit: cell-cell interactions | |
| Itga7 | Integrin subunit: cell-cell interactions | a, c |
| Myo10 | Headless myosin: roles in development and cell migration | a, c |
| Gfra1 | Receptor for GDNF: promotes neural survival and differentiation of neurons | a |
| Lmo7 | Signaling: negative feedback regulator of transforming growth factor | a, b |
| Pou2f1 | Transcription factor: prominent in development | a, c |

**Table 1b**

| Gene | Reported functional roles | Reported previously[3] |
|---|---|---|
| Grik1 | Ligand-gated ion channel subunit: kainate receptor type 1 | a |
| Prkca | Signal transduction: protein kinase c alpha | |
| Trpc3 | Ion channel: Trpc3 linked to mechanosensory transduction | |
| Scn10a | Sodium ion channel subunit: Nav1.8 linked to pain sensation in humans and mice | a |
| Scn9a | Sodium ion channel subunit: Nav1.7 linked to pain sensation in humans and mice | |
| Scn1a | Sodium ion channel subunit: Nav1.1 linked to pain sensation in humans and mice | a |
| Calca | Neuropeptide: CGRP, role in neuroinflammation and pain | |
| Tac1 | Neuropeptide: substance P, acts as a neurotransmitter/modulator, role in pain | a |
| Kcnb2 | Potassium channel subunit | a |
| Cacnb4 | Calcium channel subunit | |
| Trpm3 | Ion channel: Trpm3 linked to pain signaling in mice | |
| Oprm1 | G protein coupled receptor: mu-opioid receptor | |
| Snap25 | Synaptosome associated protein: role in neurotransmitter release | a |

[1]Upregulated after SNI (a) *Shin et al. (2019)*, (b) *Wlaschin et al. (2018)*, (c) *Cobos et al. (2018)*.
[2]Role in neural regeneration (D) *Mahar and Cavalli (2018)*.
[3]Downregulated after SNI (a) *Shin et al. (2019)*.
DOI: https://doi.org/10.7554/eLife.49679.004

in I1 neurons (*Figure 2—figure supplement 2*). By contrast the neuropeptide *Gal* was largely restricted to I2 (*Figure 2—figure supplement 2*) and selectively labeled just a subset of injured neurons in double label ISH (*Figure 2g*). Genes that best distinguish I1 and I2 were subjected to gene enrichment ontology analysis (GO analysis, *Figure 2—figure supplement 3*). Although I2 markers were enriched in genes related to pain in line with our predictions, most pathways identified by this analysis were very general. Thus, taken together our data show that axon-transection induces a remarkably consistent transcriptional transformation in the diverse types of trigeminal neurons to produce an injured neural state retaining just vestiges of the gene expression patterns that distinguish classes of uninjured neurons.

## Long-term survival and transcriptional stability of injured trigeminal neurons

To assess how gene expression evolved after SNI, we carried out sn-transcriptomic analysis using ganglia isolated 7-, 21- and 75 days post injury and used a unified method (*Butler et al., 2018*) to co-cluster the sequencing data from the uninjured control and the four IOT time-points (*Figure 3*). As expected, uninjured classes of trigeminal neuron were all distinguished in this analysis and segregated from the injured neural classes (*Figure 3a,b*). However, the most striking feature revealed by our analysis was the stability of the clustering, including all the uninjured clusters and the two classes of injured neurons (I1, I2) across the time-course (*Figure 3c*). Indeed, the most salient difference between timepoints was the number of injured neurons. At 7 days after injury, 539 from 4033 of total neurons (approx. 13.4%) grouped into I1 and I2, similar to the 13.8% observed at day 2. However, by 21 days post injury only 288 of 4046 neurons (7.1%) segregated into the injury-related clusters and 75 days after surgery this ratio had fallen further to 104 of 3634 neurons (approx 2.9%). Therefore, although the injury-related states are very long-lived, there is a progressive loss of cells with these transcriptomic features at longer times after injury.

The consistent clustering patterns of the individual timepoints in the UMap projection (*Figure 3c*) from 2 to 75 days after injury rule out the possibility that either injured or uninjured neurons slowly transition into new states during this timeframe. Moreover, analyzing gene expression in the injured state neurons clusters more thoroughly (see *Supplementary file 3*), revealed just a small number of upregulated genes changed expression level after day 2. Examples include the Lim only domain protein 7 (*Lmo7*), which was rapidly upregulated but downregulated at later time-points and two injury-related neuropeptides *Nts* and *Npy* that were more slowly upregulated (*Figure 3d*). Multicolor ISH graphically demonstrated these temporal differences in neuropeptide expression at a cellular level (*Figure 3e*) and revealed the complex pattern of their expression in injured trigeminal neurons providing a clear example of variation amongst the injured cells. However, the most significant conclusion from the time-course of transcriptomic changes following injury was that within 2 days of IOT, damaged neurons adopted a very stable injured state that was quite distinct from any undamaged class of trigeminal neurons. In contrast, mechanical sensitization and tactile allodynia usually take a week or more to fully develop after nerve injury (*Cobos et al., 2018*; *Hardt et al., 2019*; *Wlaschin et al., 2018*). Therefore, our results are consistent with recent data suggesting that central processing of somatosensory input plays a major role in this type of neuropathic pain (*Guan et al., 2016*; *Szczot et al., 2018*).

## Induction of the Atf3-transcriptional state is a physiological response to peripheral nerve damage

Nociception has an important role in guarding against immediate dangers and in teaching us to avoid risk (*Basbaum et al., 2009*; *Julius, 2013*). Similarly, trauma-induced pain probably plays a vital role in protecting animals from exacerbating injury (*Cox et al., 2006*; *Goldberg et al., 2007*). However, neuropathic pain does not serve such an evolutionarily significant purpose. Moreover, our results suggest that even if injury-related gene expression changes are required for nerve injury-related tactile allodynia (*Guan et al., 2016*; *Wlaschin et al., 2018*), the injury-related transcriptomic state develops long before pain symptoms are maximal. Therefore, we reasoned that the stereotyped transcriptional response that defines I1 and I2 likely has another and completely distinct role in somatosensation. Indeed, the presence of a small number of neurons in uninjured control animals that exhibited these gene expression profiles (*Figure 3c*) suggested that the injured state might be

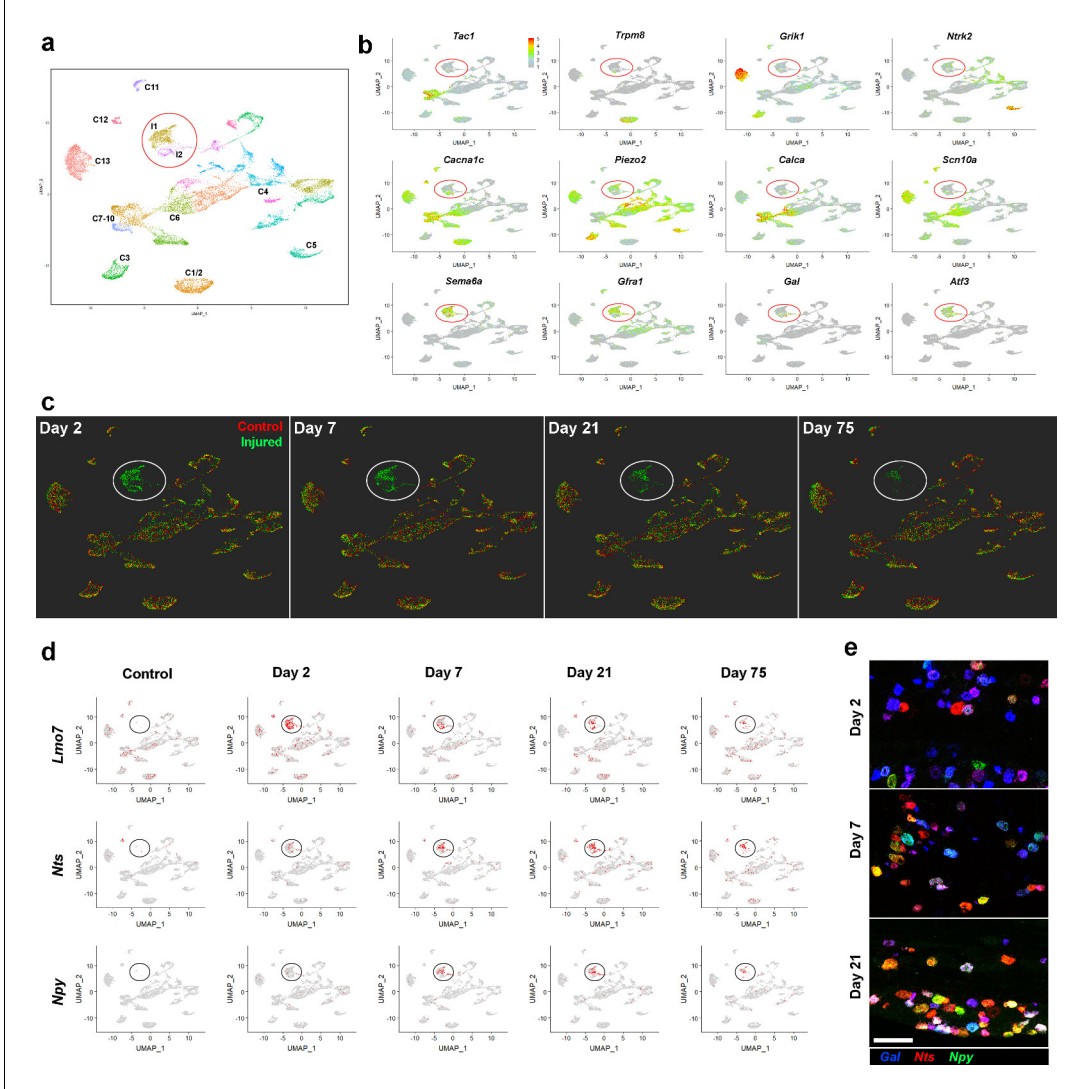

**Figure 3.** Stability of the IOT induced injured neural state. (**a**) UMap representation of sn-data: co-clustering of 20,692 trigeminal neurons from control uninjured mice and from mice at day 2, day 7, day 21, day 75 after IOT; injury-related classes I1 and I2 are circled; colors distinguish clusters identified in this analysis. (**b**) Expression of select genes (relative cellular expression indicated by the scale-bar) shows down-regulation of several key somatosensory genes in the injured neurons (circled) and upregulation of injury-related transcripts. (**c**) Cells from each of the four time-points after injury (green) are shown separately in UMap representations. For comparison the uninjured control data (red) are included in each plot. Throughout the time-course, both the injured neurons (circled) and uninjured neural classes remained stable in this representation of multidimensional space. Notably, however, the number of injured neurons decreased at 21 and 75 days. (**d**) UMap representation showing expression of three injury induced genes (gray, no expression; red, expression; injured cell-classes circled). Note that *Lmo7* shows decreased expression after Day 2, whereas expression of *Nts* and *Npy* increases at these later time-points. (**e**) Representative images of triple label ISH using probes for the injury induced neuropeptides *Gal* (blue), *Nts* (red) and *Npy* (green) expose the complex co-expression patterns of these genes after IOT and upregulation of *Nts* and *Npy* at later time-points; scale-bar = 100 μm.

DOI: https://doi.org/10.7554/eLife.49679.009

induced without axotomy and led us to investigate if modest nerve damage could trigger the transformation.

Initial support for the idea that peripheral injury induces the same transcriptomic transformation as SNI came from studying sham IOT surgery controls that we had used for ISH analysis. This surgery involves a cut to the facial skin and blunt dissection of muscle to expose but not injure the nerve and thus results in a localized but significant peripheral injury. In mice subjected to this surgery, ISH consistently revealed a small number of *Atf3* positive cells that were clustered together in the trigeminal

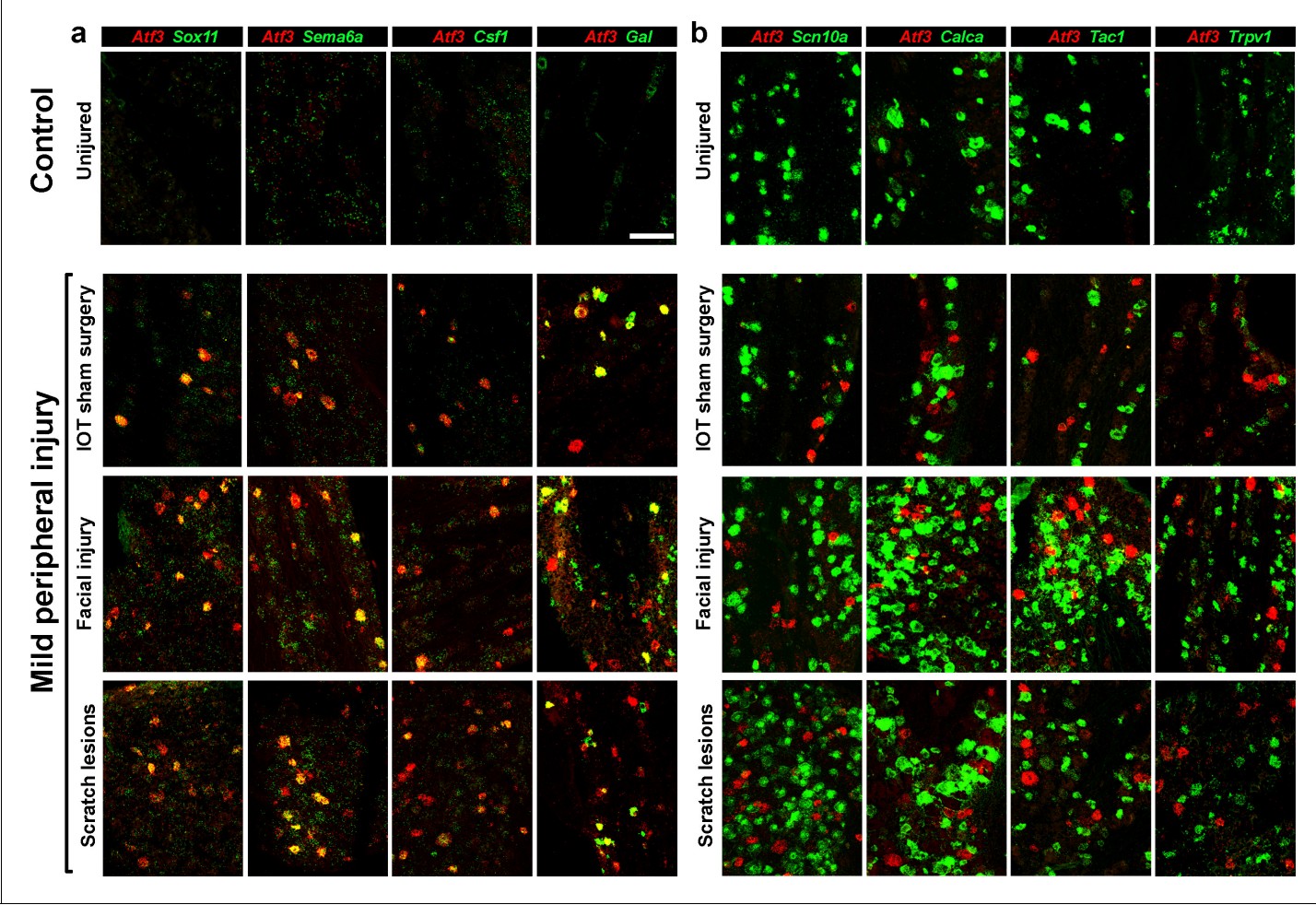

**Figure 4.** Several types of peripheral injury triggered gene expression changes paralleling those induced by IOT. Representative images of double label ISH, (*Atf3*, red) illustrate. (**a**) upregulation of other injury-related genes (green) and (**b**) down regulation of key somatosensory transcripts (green) in damaged neurons expressing *Atf3*. Uninjured control mice typically showed no *Atf3* expression (top panels). In contrast three types of mild peripheral injury, IOT sham surgery, facial injury and lesions to the head and neck resulting from scratching (lower panels) induced *Atf3*-expression in a subset of regionally localized trigeminal neurons. Time points analyzed here were 2 days after injury for IOT sham surgery and facial injury models; scale-bar = 100 μm, see *Supplementary file 1* for quantitation of data.

DOI: https://doi.org/10.7554/eLife.49679.010

ganglion (*Figure 4a*). These cells expressed several genes that are normally upregulated after nerve injury including *Sox11*, *Sema6a*, *Csf1* and *Gal* (*Figure 4a*). We next investigated whether other types of mild peripheral injury also induced *Atf3* expression. To do this, we used a minor facial injury model (a series of extensive skin incisions to the facial area that just cut through the dermis) and examined mice that exhibited a chronic itch-like phenotype where scratching had produced obvious shallow skin lesions in the head and neck area (scratch lesions). In both cases, *Atf3*-positive neurons were detected by ISH and just as in the case of sham surgery these cells also expressed other markers of nerve injury (*Figure 4a*). Thus, it appears that the same transcriptomic program (*Wlaschin et al., 2018*) is upregulated following various types of relatively minor peripheral injury as well as after transection of major branches of the trigeminal nerve. The three types of minor peripheral injury also downregulated functionally significant genes. Just as we showed after IOT (*Figure 2*), *Atf3*-positive neurons expressed only low levels of key genes including *Scn10a*, *Calca*, *Tac1* and *Trpv1* (*Figure 4b*), suggesting that the normal sensory function of these cells must be dramatically changed.

The similarity between injury models indicated that induction of the injury-related state is a normal response of somatosensory neurons to peripheral damage. However, our ISH results suggested that there might be some minor differences between models. For example, the level of expression of injury-related genes appeared more variable in animals with mild facial injury or scratch lesions (*Figure 4a*) than for those subjected to IOT (*Figure 2—figure supplement 1b*) and although *Scn10a* and *Trpv1* expression was much weaker in injured cells (*Figure 4b*), they were detectable in a larger proportion of injured cells in these models than after IOT (*Supplementary file 1*). These differences may reflect a graded response to peripheral injury. We also observed many small diameter *Gal*-positive, *Atf3*-negative cells after these types of peripheral injury (*Figure 4a*) that were not detected following nerve transection (*Figure 2a*). On the one hand this may indicate that damage to nociceptors can induce distinct patterns of gene expression, but on the other, it may be that local effects of injury can influence gene expression in undamaged neurons. To explore these issues and to better and quantitatively define the extent to which mild peripheral injury mimics full axotomy, we carried out sn-transcriptomic analysis of trigeminal neurons from mice with facial injury and itch lesions (*Figure 5*, *Figure 5—figure supplement 1*). We did not carry out sn-sequence analysis for mice subjected to sham IOT surgery because of its localized and more severe nature.

Initial clustering of the two datasets from mice with facial injury or scratch lesions (*Figure 5—figure supplement 1*) revealed that most neurons grouped into clusters resembling those identified from uninjured control mice (*Figure 1*). However, in both cases an additional class of cells clustered separately (*Figure 5—figure supplement 1a*) and expressed several markers of damaged trigeminal neurons (*Figure 5—figure supplement 1b*). The injury-related cells accounted for approximately 1.3% and 2.7% of total neurons in the facial injury and scratch lesion models, respectively. To better assess how these neurons were related to the I1 and I2 classes that result from IOT, we co-clustered all data (*Figure 5a*) and then compared the two peripheral injury models with effects of nerve transection (*Figure 5b,c*). This approach makes use of information about the cellular expression level of many genes both to define and display clusters (*Butler et al., 2018*) and thus is highly quantitative. Remarkably, *Atf3*-positive neurons from both minor injury models were grouped with I1 and I2 neurons from IOT mice (*Figure 5b*). There were small differences, for example, in scratch lesion mice, most damaged neurons were in the I2 class with several likely related to normal C3 neurons (*Figure 5b*). Such variation may reflect the type of injury and/or the innervation of the injured sites by a slightly different distribution of neural-classes from those damaged in the IOT model. One other difference that was evident for both minor injury models was low-level upregulation of *Gal* in neurons outside the I1 and I2 classes (*Figure 5c*) in keeping with ISH results (*Figure 4a*). However, these data and additional gene expression analysis (*Figure 5—figure supplement 1c* and *Figure 5—figure supplement 2*) confirm great similarity but not complete identity between the transcriptomic profiles of neurons damaged in peripheral injury models and after nerve transection.

## Genetically marking the injured state neurons reveals their long-term fate

Since induction of an injured neural state appears to be a physiologically relevant somatosensory neural response to a variety of peripheral injuries, we anticipated that it normally would play a role in the recovery process and therefore, that changes in gene expression should often be reversible. Indeed, after facial cuts had healed, ISH localization anecdotally indicated decreased expression of injury-related transcripts. Nonetheless, even three weeks after injury and thus long after healing of the skin, a few *Atf3*-positive cells were still observed in the trigeminal ganglion. The time-dependent decrease in proportion of neurons with an injury-related transcriptional profile after IOT (*Figure 3*) and the absence of new transcriptomic clusters (*Figure 3*) also indicated that recovery might occur even after nerve transection. However, in both cases it is equally possible that injured neurons progressively die, and indeed significant cell death after SNI has been well characterized (*Himes and Tessler, 1989*; *Vestergaard et al., 1997*).

Therefore, to explicitly test if neurons can recover after entering the injured state, we generated *Atf3-IRES-Cre* knockin mice, where damaged neurons could be permanently marked using Cre-recombinase activity (see *Figure 6—figure supplement 1* for details). We then used a viral strategy to transduce a subset of trigeminal neurons by intraperitoneal injection of a *Cre*-dependent adeno associated virus (AAV9-*CAG-flex-tdTomato*) in neonate knockin mice (*Szczot et al., 2018*). This approach transduces many somatosensory neurons but avoids complications from developmental

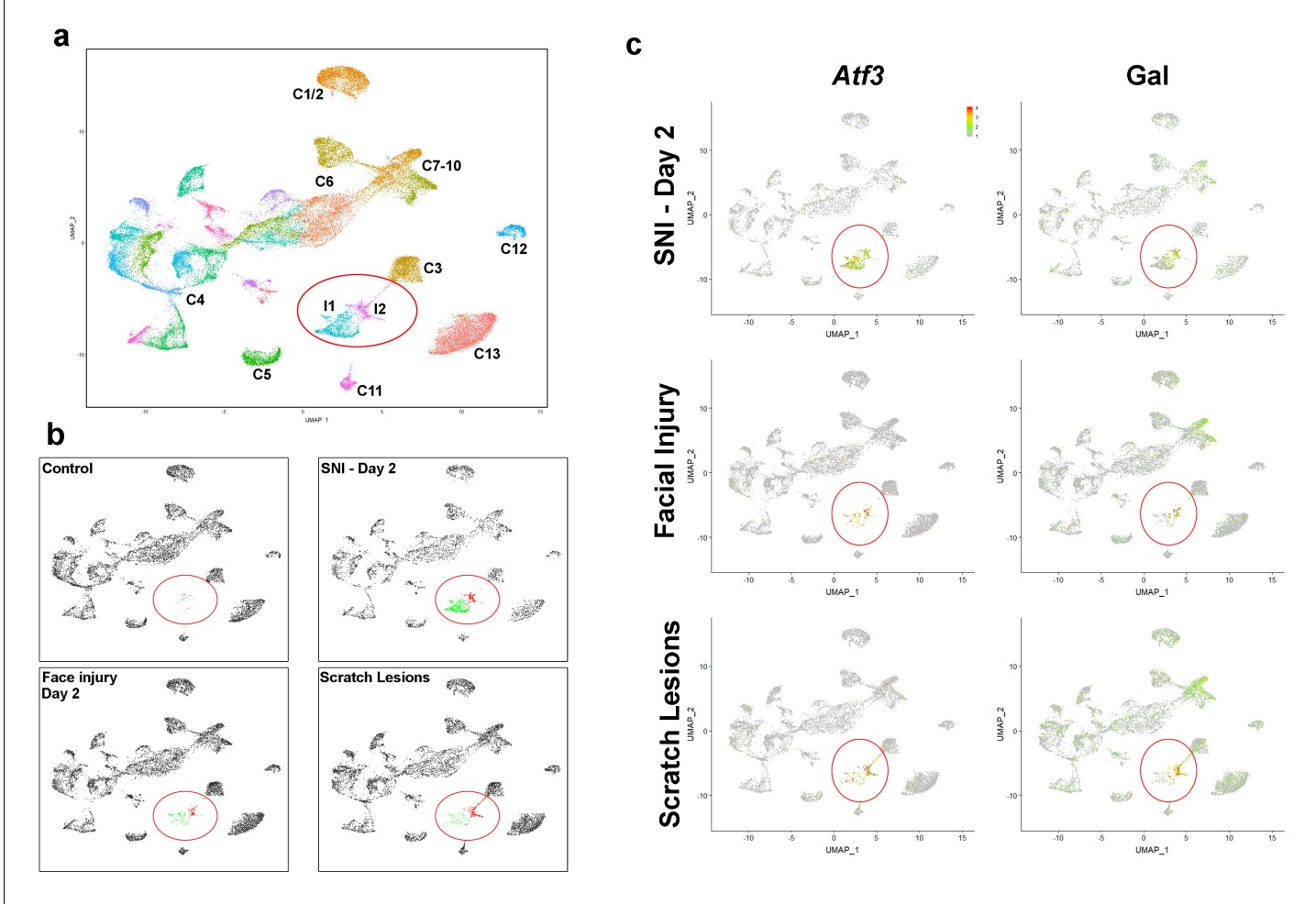

**Figure 5.** Neurons damaged by facial injury or scratching induce the same transcriptomic states as SNI. (**a**) Co-clustering of sn-data (40,359 trigeminal neurons from a combination of control uninjured mice; mice at day 2, day 7, day 21, day 75 after SNI; facial injury-day 2, and scratch lesion models); each cluster is assigned a distinct color and injury-related classes I1 and I2 are circled. The designations (C1–C13) are based on earlier single cell data (*Nguyen et al., 2017*) and do not distinguish between the new classes of cells identified here (see *Supplementary file 4* for detailed description of trigeminal neural classes). (**b**) Separate UMap plots of the data from the control, SNI-day 2, face injury and scratch injury models in the combined clustering; uninjured cells, gray, I1 cells, green; I2 cells, red; injured classes are circled. Note that both types of peripheral injury produce the same classes of injured cells as SNI; in both cases the proportion of I2 cells was higher than after nerve transection. Uninjured neurons cluster just as in control animals for all three injury models. (**c**) Injury induced transcripts *Atf3* and *Gal* were prominently expressed by injured neural classes (circled) in all three models; relative expression colored as indicated by scale-bar. However, peripheral injury also up-regulated expression of *Gal* in C7-10 cells in line with ISH results (*Figure 3*).

DOI: https://doi.org/10.7554/eLife.49679.011

The following figure supplements are available for figure 5:

**Figure supplement 1.** Peripheral injury induces gene expression changes matching those triggered by cutting major nerve branches.
DOI: https://doi.org/10.7554/eLife.49679.012

**Figure supplement 2.** Additional quantitation of genes up- and downregulated in injured trigeminal neurons Dot plot representation of gene expression in the cluster analysis shown in *Figure 5*.
DOI: https://doi.org/10.7554/eLife.49679.013

**Figure supplement 3.** GO-analysis of genes that are up- and downregulated in injured neurons.
DOI: https://doi.org/10.7554/eLife.49679.014

expression of *Atf3*. As expected, adult animals, not subjected to nerve injury exhibited only very sparse expression of *Atf3* and no detectable *tdTomato* in trigeminal neurons (*Figure 6a*). In contrast, after IOT, prominent labeling of trigeminal neurons with *tdTomato* was observed (*Figure 6b–d*).

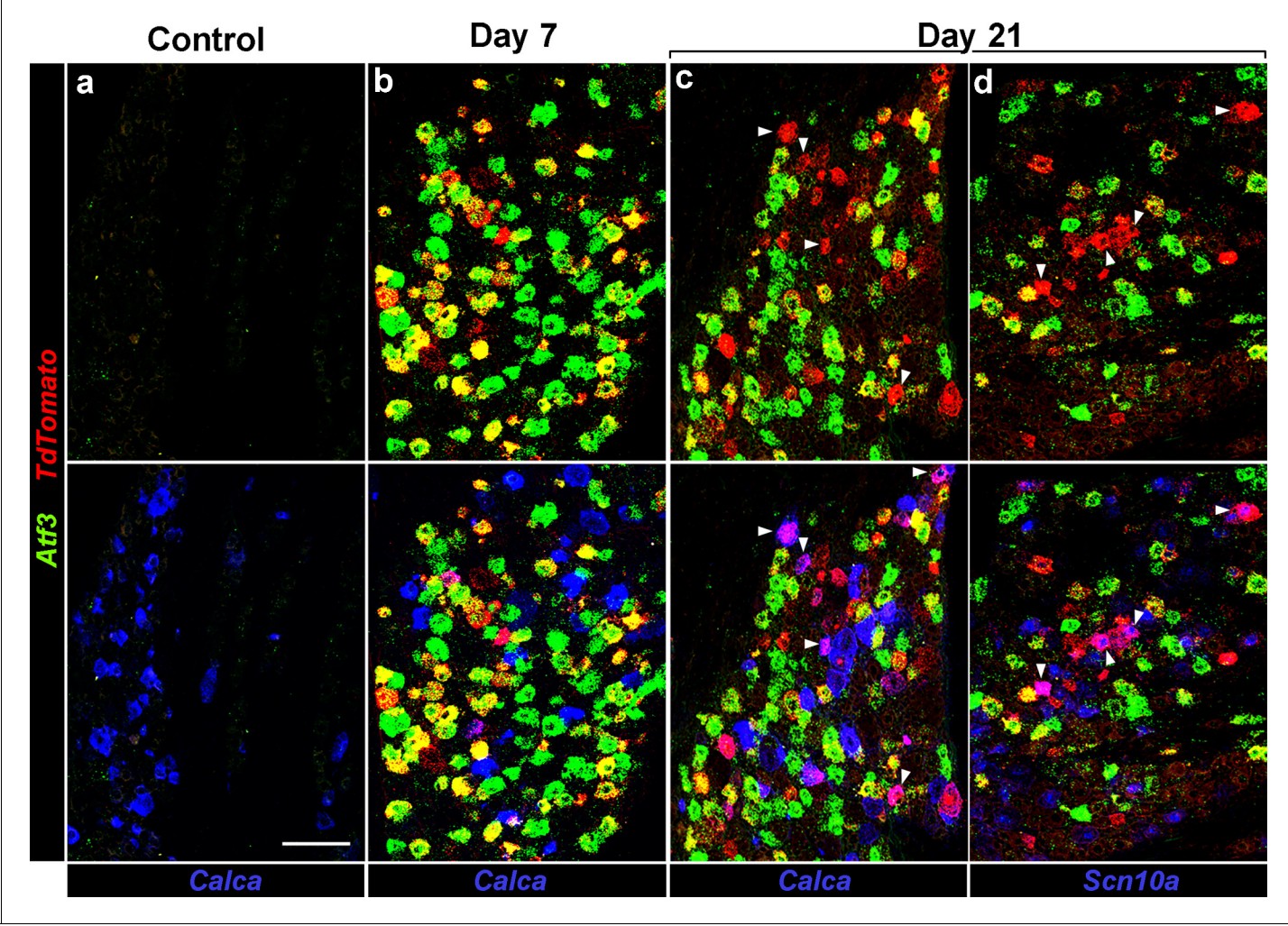

**Figure 6.** Injured neurons revert to an uninjured transcriptional state. Representative triple label ISH showing sections from *Atf3-IRES-Cre* mice where a subset of peripheral neurons has been transduced by AAV9-*CAG-flex-tdTomato*; (**a**) Before injury (Control), (**b**) 7 days and (**c,d**) 21 days after IOT. Upper panels show staining for *Atf3* (green) and *tdTomato* (red): before injury no positive cells were detected; day 7 most *tdTomato* positive neurons also expressed *Atf3*; day 21 about half the *tdTomato* positive neurons were *Atf3*-negative. Note because of the viral approach not all *Atf3*-positive cells express *tdTomato*. Lower panels show the same images but include expression of *Calca* or *Scn10a* (blue) demonstrating that these genes are rarely co-expressed with *Atf3* but are regularly found in cells labeled by *tdTomato* but not *Atf3*; arrowheads (**c,d**) point to *tdTomato*-positive cells which also express (**c**) *Calca* or (**d**) *Scn10a*. Scale-bar = 100 µm; see *Supplementary file 1* for quantitation and statistical analysis.

DOI: https://doi.org/10.7554/eLife.49679.015

The following figure supplement is available for figure 6:

**Figure supplement 1.** Generation and characterization of Atf3-IRES-Cre mice.

DOI: https://doi.org/10.7554/eLife.49679.016

Note that the viral labeling strategy targets about 30–50% of trigeminal neurons meaning that only a proportion of the *Atf3-Cre* neurons (green) activate *tdTomato* expression (red).

We reasoned that if the gradual loss of damaged neurons (approx. 50% after 21 days) exclusively reflected cell death (*Himes and Tessler, 1989*; *Vestergaard et al., 1997*) then *tdTomato* labeled neurons would always express *Atf3* (and their number would decrease with time). However, if they also reverted to an uninjured transcriptional state, these cells would continue to express *tdTomato* even though they were no longer positive for *Atf3*. Our results (*Figure 6b–d*) show that the fraction of *Atf3* negative, *tdTomato*-positive (red-only) neurons rose with time after injury indicating a progressive recovery occurs. Quantitation of data revealed that the percentage of *tdTomato*-neurons

that were *Atf3*-negative rose from 11.0 ± 5.3% at day-7 to 42.7 ± 14.7% at day-21 (see *Supplementary file 1* for details and statistics). This change parallels the decrease in number of I1 and I2 neurons identified by sn-transcriptomics (*Figure 3*), demonstrating that even after IOT many of the damaged neurons recover rather than die. These cells must reacquire a normal trigeminal neuron gene expression profile since no new classes of trigeminal neuron were detected in corresponding sn-analysis (*Figure 3*). Further support for this conclusion comes from examination of genes that are downregulated in *Atf3*-positive cells after injury. For example, *Calca* and *Scn10a* were very rarely co-expressed with *Atf3* (*Figure 6* lower panels) but were detected in a large subset (approx. 40%, see *Supplementary file 1* for quantitation and statistics) of the recovered neurons (*Figure 6c,d*). In the future, it will be important to assess whether neurons return to their own original transcriptional state after recovery from injury. However, the residual traces of original neural class that were retained amongst damaged neurons (*Figure 2—figure supplement 2*) make this the most likely scenario. It will also be interesting to see if some neural classes exhibit slower recovery and if the severity of the injury model makes a difference in this regard.

## Concluding remarks

Our results expose four key features that suggest the gene expression changes linked to SNI and the development of tactile allodynia have a much broader role as a neural response to many types of peripheral injury. First, we demonstrated that a stereotyped injured neural state is not only induced by nerve transection but also by other far less significant types of trauma. Minimal injury such as self-inflicted scratch lesions or cuts through the dermis but not deeper layers of the skin induced a very similar change in neural gene-expression profile to nerve transection. In part, this may have been missed previously because studies of nerve injury typically do not examine such minor trauma, but the power of looking at gene-expression at the single-cell level was also important. Using ISH, we showed that sham IOT surgery (another minor injury model that is less extensive) also induced injury-related transcripts in a few neurons. Second, many of the genes that are upregulated by injury have roles in tissue remodeling, neural development, and axon guidance (see *Table 1a* and *Figure 5—figure supplement 3*). These genes probably contribute to restoration of normal peripheral innervation of tissue during wound healing (*Shin et al., 2019*). Thirdly, the injury response involves downregulation of genes encoding receptors, ion channels and neuropeptides that normally play major roles in sensory detection and signaling (see *Table 1b* and *Figure 5—figure supplement 1*). We speculate that this widespread transcriptomic silencing of functionally relevant molecules may help prevent damaged neurons from conveying aberrant sensory input to central targets. Fourthly, genetic marking of damaged cells using a new *Atf3-IRES-Cre* line (*Figure 5*) and sn-transcriptomic data (*Figures 1* and *2*) show that the injury-induced state, although potentially long-lived, is transient and likely to be reversible. Thus, rather than simply being a predictor of morbidity after serious nerve damage, we surmise that the transcriptomic transformation induced by axotomy is in fact a standard response to many types of peripheral trauma probably serving as an important driver for restoration of sensation and providing a mechanism for modulating sensory input from sites of injury.

Single cell sequencing has vastly expanded the definition of neural class, with the general assumption that distinct transcriptomic classes are fundamentally different types of cell (*Li et al., 2016*; *Nguyen et al., 2017*; *Tasic et al., 2018*; *Usoskin et al., 2015*; *Zeisel et al., 2018*). Here, sn-sequencing defined a dozen new types of large diameter trigeminal neurons (*Supplementary file 2*) that may have select functions in various types of mechanosensitive response. Notably, however, we also demonstrated that the transcriptomic diversity that defines 27 distinct classes of uninjured trigeminal neurons is almost completely lost within 2 days of nerve injury. This would be remarkable enough if the transformation required gross neural damage but is all the more so because relatively minor peripheral injury also triggers this abrupt transition. Working out the signaling mechanisms that preserve normal neural gene expression and cause its collapse after injury will be of great interest. However, the simple fact that the different transcriptomic classes can all be rapidly transformed to a much more uniform state (and then reappear on recovery from injury) exposes great transcriptional plasticity amongst somatosensory neurons.

What is the role of the injured state in triggering neuropathic pain? Here we demonstrate that in this experimental model the injury-related gene expression program is essentially complete two days after IOT and thus precedes the peak development of pain (*Cobos et al., 2018*; *Hardt et al.,*

*2019*; *Wlaschin et al., 2018*). Thus, it is likely that symptoms are induced well downstream of gene-regulation in injured cells in keeping with recent studies suggesting that neuropathic pain and mechanical allodynia primarily involves modulation of central processing rather than altered sensory input (*Guan et al., 2016*; *Szczot et al., 2018*). Further support for this comes from the fact that uninjured neurons are unchanged at a transcriptomic level after SNI. It is interesting that more minor injuries can induce a very similar pattern of gene expression, raising the possibility that long-term peripheral trauma and delayed recovery might have some potential to trigger similar types of neuropathy to more frank nerve injury.

Finally, we note that the type of nerve damage caused by transection makes it unlikely that precise reinnervation of target sites can be required for neurons to initiate the transition back to normal function. Indeed, it is possible that aberrant signaling from neurons with inappropriate peripheral endings could contribute to some types of pain and sensory disturbances associated with severe injuries. In the future, we expect that genetic approaches using the *Atf3-Cre* driver to ablate and silence damaged neurons will help address these issues and reveal if and when these injured neurons are required for development (and perhaps resolution) of various types of pain.

# Materials and methods

## Key resources table

| Reagent type (species) or resource | Designation | Source or reference | Identifiers | Additional information |
|---|---|---|---|---|
| Gene (*Mus musculus*) | Atf3 | NA | ENSMUS G00000026628 | |
| Strain, strain background (*Mus musculus*) | C57BL/6 | Charles River | Strain code: 027 | |
| Genetic reagent (*Mus musculus*) | Atf3-IRES-Cre | This paper | | Knockin mouse; details *Figure 6— figure supplement 1* |
| Genetic reagent (*Mus musculus*) | TetO-mCherry-2A-Gnaq* | This paper | | Random insertion transgene (See Materials and methods) |
| Genetic reagent (*Mus musculus*) | Fos-tTA | The Jackson Laboratory | JAX: 018306 | *Reijmers et al., 2007* |
| Genetic reagent (Adeno-associated virus) | AAV9-CAG-FLEX-tdTomato-WPRE | Addgene; Oh et al., 2014 | Addgene: 51503-AAV9 | $2.1 \times 1013$ GC/µl |
| Antibody | Anti-NeuN (rabbit polyclonal) | Millipore | Cat#ABN78; RRID:AB_10807945 | (1:2000) |
| Sequence-based reagent | crRNA | Dharmacon Inc. | Edit-R Modified Synthetic crRNA | Target sequence: GCAGAAGTGTCTACCTTGAT |
| Peptide, recombinant protein | Cas9 | PNA Bio Inc. | CP01 | |
| Commercial assay or kit | RNAscope multiplex fluorescent development kit | Advanced Cell Diagnostics | ACD: 320851 | |
| Commercial assay or kit | RNAscope probe-Mm-S100b | Advanced Cell Diagnostics | ACD: 431731 | |
| Commercial assay or kit | RNAscope probe-Mm-Slit2 | Advanced Cell Diagnostics | ACD: 449691 | |
| Commercial assay or kit | RNAscope probe-Mm-Ntng1 | Advanced Cell Diagnostics | ACD: 488871 | |
| Commercial assay or kit | RNAscope probe-Mm-Rgs6 | Advanced Cell Diagnostics | ACD: 521211 | |
| Commercial assay or kit | RNAscope probe-Mm-Cacna1h | Advanced Cell Diagnostics | ACD: 459751 | |

*Continued on next page*

*Continued*

| Reagent type (species) or resource | Designation | Source or reference | Identifiers | Additional information |
|---|---|---|---|---|
| Commercial assay or kit | RNAscope probe-Mm-Scn5a | Advanced Cell Diagnostics | ACD: 429881 | |
| Commercial assay or kit | RNAscope probe-Mm-Htr3a | Advanced Cell Diagnostics | ACD: 411141 | |
| Commercial assay or kit | RNAscope probe-Mm-Mrgprd | Advanced Cell Diagnostics | ACD: 417921 | |
| Commercial assay or kit | RNAscope probe-Mm-Scn10a | Advanced Cell Diagnostics | ACD: 426011 | |
| Commercial assay or kit | RNAscope probe-Mm-Atf3 | Advanced Cell Diagnostics | ACD: 426891 | |
| Commercial assay or kit | RNAscope probe-Mm-TrpV1 | Advanced Cell Diagnostics | ACD: 313331 | |
| Commercial assay or kit | RNAscope probe-Mm-Tac1 | Advanced Cell Diagnostics | ACD: 410351 | |
| commercial assay or kit | RNAscope probe-Mm-Calca | Advanced Cell Diagnostics | ACD: 420361 | |
| Commercial assay or kit | RNAscope probe-Mm-Grik1 | Advanced Cell Diagnostics | ACD: 438771 | |
| Commercial assay or kit | RNAscope probe-Mm-Piezo2 | Advanced Cell Diagnostics | ACD: 400191 | |
| Commercial assay or kit | RNAscope probe-Mm-Gal | Advanced Cell Diagnostics | ACD: 400961 | |
| Commercial assay or kit | RNAscope probe-Mm-Nts | Advanced Cell Diagnostics | ACD: 420441 | |
| Commercial assay or kit | RNAscope probe-Mm-Npy | Advanced Cell Diagnostics | ACD: 313321 | |
| Commercial assay or kit | RNAscope probe-Mm-Sox11 | Advanced Cell Diagnostics | ACD: 440811 | |
| Commercial assay or kit | RNAscope probe-Mm-Sema6a | Advanced Cell Diagnostics | ACD: 508101 | |
| Commercial assay or kit | RNAscope probe-Mm-Csf1 | Advanced Cell Diagnostics | ACD: 315621 | |
| Commercial assay or kit | RNAscope probe-TdTomato | Advanced Cell Diagnostics | ACD: 317041 | |
| Commercial assay or kit | RNAscope probe-Fos | Advanced Cell Diagnostics | ACD: 316921 | |
| Commercial assay or kit | RNAscope probe-Egr1 | Advanced Cell Diagnostics | ACD: 423371 | |
| Commercial assay or kit | Chromium single cell 3' reagent kit (v2) | 10X Genomics | Cat# 120237 | |
| Commercial assay or kit | Chromium i7 Multiplex Kit, 96 rxns | 10x Genomics | Cat# 120262 | |
| Commercial assay or kit | Chromium Single Cell A Chip Kit | 10x Genomics | Cat# 120236 | |
| Software, algorithm | Seurat | *Butler et al., 2018*; *Stuart et al., 2018* | RRID:SCR_016341 | https://satijalab.org/ |
| Software, algorithm | CellRanger | 10x Genomics | | |
| Software, algorithm | Drop-seq | McCarroll Lab | | http://mccarrolllab.org/dropseq/ |
| Software, algorithm | R | R Project for Statistical Computing | RRID:SCR_001905 | http://www.r-project.org/ |

*Continued on next page*

*Continued*

| Reagent type (species) or resource | Designation | Source or reference | Identifiers | Additional information |
|---|---|---|---|---|
| Software, algorithm | R Studio | R Studio | rstudio.com | |
| Software, algorithm | Prism v8 | GraphPad | RRID:SCR_002798 | |
| Software, algorithm | PhotoShop CC | Adobe | RRID:SCR_014199 | |
| Other | RNAlater | ThermoFisher | Cat# AM7021 | |
| Other | Glass dounce homogenizer | Fisher Scientific | Cat# 357538 | |
| Other | 40 µm cell strainer | ThermoFisher | Cat# 08-771-1 | |
| Other | SUPERaseIn RNase Inhibitor | ThermoFisher | Cat# AM2696 | 0.2 U/ml |
| Other | Anti-rabbit IgG microbeads | Miltenyi Biotec | Cat# 130-048-602 | |
| Other | LS columns | Miltenyi Biotec | Cat# 130-042-401 | |
| Other | MACS MultiStand | Miltenyi Biotec | Cat# 130-042-303 | |
| Other | MidiMACS Separator | Miltenyi Biotec | Cat# 130-042-302 | |
| Other | Ultra-Turrax T10 | Laboratory Supply Network, Inc. | IKA:3737001 | |
| Other | Dispersing element S10N-5G for Ultra-Turrax T10 | Laboratory Supply Network, Inc. | IKA:3304000 | |

## Mice, surgery and viral transduction

Animal experiments were carried out in strict accordance with the US National Institutes of Health (NIH) guidelines for the care and use of laboratory animals and were approved by the NIDCR ACUC. Male and female mice were used for all experiments but were not analyzed separately; animals were assigned to groups without randomization. Mice were C57BL/6NCrl except where specified and were 6 weeks or older at the time of surgery. IOT was carried out as previously described (*Xu et al., 2008*) and cut the superficial three branches of the infraorbital (trigeminal) nerve, which were not ligated. Sham IOT surgery exposed these branches, but the nerve was not cut or damaged, while facial injury involved a series of 3–5, approx. 0.6 cm cuts to the facial skin but did not penetrate deeper tissues; mice were used 2 days post injury in each of these mild injury models. *Atf3-IRES-Cre* mice were generated by homologous recombination in C57BL/6J mouse zygotes using clustered regularly interspersed short palindromic repeats (CRISPR) associated protein 9 (Cas9) to assist targeting (*Wu et al., 2018*) see *Figure 6—figure supplement 1* for details and characterization of the line. Intraperitoneal injection of AAV9-CAG-FLEX-tdTomato-WPRE was carried out in heterozygous neonate animals (*Szczot et al., 2018*). A standard transgenic line (random insertion in the genome) with a *tetO* repeat upstream of a *mCherry-2A-Gnaq* mutant allele encoding constitutively active Gqα was generated by pronuclear injection. The constitutively active Gqα was engineered by substituting the glutamine (CAA) at position 209 with a leucine (CTA). The mice with scratch lesions were a double heterozygous transgenic strain where a TetTag, *Fos-tTA* (*Reijmers et al., 2007*) drove expression of this *tetO-mCherry-2A-Gnaq* mutant. These mice invariably developed an itch-like phenotype, characterized by excessive scratching at 5–8 weeks of age that rapidly resulted in development of skin lesions. As expected, feeding the mice a doxycycline containing diet prevented the development of this phenotype. However, no expression of *mCherry* was detected in somatosensory neurons either before or after mice began to exhibit itch-like behavior; thus, the cause of scratching remains unclear. For the studies reported here, mice with visible lesions in the head and neck area were selected.

## Isolation of trigeminal neural nuclei

Trigeminal ganglia were harvested and incubated overnight in RNAlater (ThermoFisher, Cat# AM7021) at 4°C. Excess RNAlater was removed, ganglia were frozen on dry ice and stored at −80°C. To isolate nuclei, we adapted methods described previously (*Sathyamurthy et al., 2018*) to the

trigeminal ganglion. To generate sufficient nuclei for capture and sequencing, we pooled ganglia from several animals (males and females). For initial Dropseq capture (a subset of the uninjured controls only) we used 16 mice; 10X Chromium-based sequencing used ganglia from 10 animals in all cases except the 75 days post IOT, which was from six mice. Ganglia were homogenized in a Dounce homogenizer (Fisher Scientific, Cat# 357538); five strokes with the 'loose' pestle (A) and 15 strokes with the 'tight' pestle (B) in 1 ml of freshly prepared ice-cold homogenization buffer (250 mM sucrose, 25 mM KCl, 5 mM MgCl$_2$, 10 mM Tris, pH 8.0, 1 µM DTT, 0.1% Triton X-100 (v/v). The homogenate was filtered through a 40 µm cell strainer (ThermoFisher, cat# 08-771-1), transferred to low bind microfuge tubes (Sorenson BioScience, cat# 11700) and centrifuged at 800 g for 8 mins at 4°C. The supernatant was removed, the pellet gently resuspended in 500 µl of PBS with 1% BSA and SUPERaseIn RNase Inhibitor (0.2 U/µl; ThermoFisher, Cat#AM2696) and was incubated on ice for 10 min.

Neuronal nuclei selection was performed by incubating the sample with a rabbit polyclonal anti-NeuN antibody (Millipore, cat#ABN78) at 1:2000 with rotation at 4°C for 30 min. The sample was then washed with 500 µl of PBS with 1% BSA and SUPERaseIn RNase Inhibitor and centrifuged at 800 g for 8 mins at 4°C. The resulting pellet was resuspended in 80 µl of PBS, 0.5% BSA, 2 mM EDTA. 20 µl of anti-rabbit IgG microbeads (Miltenyi biotec, cat# 130-048-602) were added to the sample followed by a 20 min incubation at 4°C. Nuclei with attached microbeads were isolated using LS column (Miltenyi Biotec, cat# 130-042-401) according to the manufacturer's instruction. The neural nuclei enriched eluate was centrifuged at 500 g for 10 min, 4°C. The supernatant was discarded, and the pellet was resuspended in 1.5 ml of PBS with 1% BSA. To disrupt clumped nuclei, the sample was homogenized on ice with an Ultra-Turrax homogenizer (setting 1) for 45 secs. An aliquot was then stained with trypan blue and the nuclei were counted using a hemocytometer. The nuclei were pelleted at 800 g, 8 mins at 4°C and resuspended in an appropriate volume for Dropseq or 10X Chromium capture. A second count was performed to confirm nuclei concentration and for visual inspection of nuclei quality.

## Droplet capture of single nuclei, cDNA sequencing and data analysis

Dropseq capture (for a subset of control data only) and library generation were performed as described previously (*Nguyen et al., 2017*) with the following modification: the concentration of sarkosyl in the lysis buffer was doubled to decrease the droplet size. 10X Chromium capture (control and all injury data) and library generation were performed according to manufacturer's instructions using v2 chemistry kit. Next generation sequencing was performed using Illumina sequencers. Dropseq data were mapped to the transcriptome as described previously (*Nguyen et al., 2017*) except that the gene structure was modified to include intronic sequence (pre-mRNA modified mouse genome mm10). 10X Chromium data were mapped using CellRanger and the same pre-mRNA modified mouse genome (mm10). Data analysis used the Seurat V2 and V3 packages developed by the Satija lab and followed standard procedures (*Butler et al., 2018*; *Stuart et al., 2018*). For sn-RNA sequencing experiments cell filtering was performed as follows: outliers were identified and removed based on number of expressed genes and mitochondrial proportion as is standard practice in single cell transcriptomic analysis. In addition, all data were clustered using standard methods, any residual non-neuronal cell clusters were identified by their gene expression profiles: clusters not expressing somatosensory genes like *Scn9a*, *Scn10a*, *Piezo2*, etc. but instead expressing highly elevated levels of markers of non-neuronal cells including *Prp1*, *Mbp*, *Apod*, *Apoe* were tagged as non-neuronal and were removed prior to the clustering reported here. Outside single cell experiments no data were excluded. Reclustering data excluding specific genes was achieved by analyzing data for significantly up- or down-regulated transcripts and removal of these genes from the list of variable genes used in principle component selection prior to standard clustering. Numbers of sn-transcriptomes analyzed were typical for this type of experiment. GO-analysis was performed using the Gene Ontology Resource (*Ashburner et al., 2000*; *The Gene Ontology Consortium, 2018*).

## In situ hybridization

Trigeminal ganglia from mice aged 6 weeks and older were harvested and fresh frozen in OCT (Tissue-Tek). 10 µm sections were used for multi-color in situ hybridization using RNAscope Multiplex Fluorescent Assay (Advanced Cell Diagnostics) according to the manufacturer's instructions.

Confocal images were acquired with a Nikon C2 Eclipse Ti (Nikon) at 1 μm optical section. All images are maximum projection (collapsed) stacks of 10 individual optical sections; consistency of staining was assessed using multiple sections from at least three mice as is considered standard; images were processed using Adobe Photoshop CC to adjust brightness, contrast and set channel color for display. Because expression of *Atf3* was strongly clustered in the trigeminal ganglion of IOT, sham surgery and facial injury models, images shown for controls and all injury models are of this region. Cell counts were determined from images of multiple sections from three animals. Total neuronal counts were determined using ISH for tubulin beta type 3 (*Tubb3*), injured neurons were identified using ISH for *Atf3* and recovered neurons (*Figure 6*) were defined as *tdTomato*-positive, *Atf3*-negative. The fraction of these populations positive for a specific marker gene was determined for each section in a series; the reported expression levels (*Supplementary file 1*) are the mean ± standard deviation across sections because biological variation includes regional differences in expression of genes in the trigeminal ganglion. Sections were collected from three mice for all quantitation. Statistical analysis of ISH data was carried out using Graph Pad Prism and is reported in *Supplementary file 1*. For simple comparisons we used Welch's t-test since it makes no assumption of equal variance; multiple comparisons used a one-way ANOVA with Dunnett's T3 post hoc testing since we were only testing one dataset against two others.

## Data availability statement

All sequence data have been deposited in GEO (Accession number: GSE131272). Other data that support the findings of this study are available from the corresponding author upon request. There are no restrictions on data availability.

## Acknowledgements

This work was supported by the Intramural Program of the National Institutes of Health, National Institute of Dental and Craniofacial Research (NJPR) and National Institute of Child Health and Human Development (CELP) and was also supported by the NIH DDIR Innovation Fund (NJPR, CELP). We utilized the computational resources of the NIH HPC Biowulf cluster. (http://hpc.nih.gov); 10X genomics capture and all DNA sequencing were carried out by the NIDCR Genomics and Computational Biology Core. We are also grateful to Dr. Jim Pickel and the NIMH transgenic core for help generating knockin mice and to Drs. Alex Chesler and Mark Hoon as well as members of our groups for valuable input.

## Additional information

### Funding

| Funder | Grant reference number | Author |
| --- | --- | --- |
| National Institute of Dental and Craniofacial Research | ZIA-DE000561 | Nicholas Ryba |
| Eunice Kennedy Shriver National Institute of Child Health and Human Development | ZIA-HD008966 | Claire E Le Pichon |
| National Institutes of Health | DDIR Innovation Awards Program | Nicholas Ryba Claire E Le Pichon |

The funders had no role in study design, data collection and interpretation, or the decision to submit the work for publication.

### Author contributions

Minh Q Nguyen, Conceptualization, Resources, Data curation, Formal analysis, Investigation, Methodology, Writing—review and editing; Claire E Le Pichon, Conceptualization, Resources, Writing—review and editing; Nicholas Ryba, Conceptualization, Resources, Formal analysis, Supervision, Funding acquisition, Investigation, Methodology, Writing—original draft, Writing—review and editing

## Author ORCIDs
Claire E Le Pichon http://orcid.org/0000-0002-9274-3615
Nicholas Ryba https://orcid.org/0000-0002-2060-8393

## Ethics
Animal experimentation: Animal experiments were carried out in strict accordance with the US National Institutes of Health (NIH) guidelines for the care and use of laboratory animals and were approved by the NIDCR ACUC protocols #17-847 and 18-868.

## Decision letter and Author response
Decision letter https://doi.org/10.7554/eLife.49679.027
Author response https://doi.org/10.7554/eLife.49679.028

## Additional files

### Supplementary files
• Supplementary file 1. Quantitation and statistical analysis of ISH data. Quantitation was carried out by counting positive neurons in multilabel ISH images of trigeminal ganglion sections from 3 mice of a given genotype and injury model. Expression ratios were calculated for each section and used to determine mean and standard deviation. Significant differences in the proportion of a given population of neurons expressing a marker were determined using Welch's t-test (one-tailed); p-values>0.0001 are reported explicitly.
DOI: https://doi.org/10.7554/eLife.49679.017

• Supplementary file 2. Genes that differentiate I1 from I2. The clustering in *Figure 5* was used as a framework to search for markers for the I1 versus I2 clusters using the FindMarkers function and the default Wilcoxon rank sum test in Seurat; a minimum criterion for inclusion was expression in 20% of cells. Markers for I1 and I2 are displayed in separate sheets (click through tabs at the bottom of the spreadsheet).
DOI: https://doi.org/10.7554/eLife.49679.018

• Supplementary file 3. Markers for each injury model. The clustering in *Figure 5* was used as a framework to search for markers in the I1+I2 clusters versus all other clusters for each injury model using the FindMarkers function and the default Wilcoxon rank sum test in Seurat; expression in more than 20% of injured or noninjured cells was a criterion for inclusion. Genes are displayed in separate sheets labeled according to model and time (click through tabs at the bottom of the spreadsheet). It should be noted that even after removal of injury-affected genes from the variable genes used in clustering, the injury-related classes were still resolved (*Figure 2*). Therefore, although these are the most prominent effects of a particular injury on trigeminal neural gene expression the effects of these types of injury are likely to be far more extensive.
DOI: https://doi.org/10.7554/eLife.49679.019

• Supplementary file 4. Markers for each of the trigeminal neural classes. Markers for the classes identified in the clustering (*Figure 5*) were determined using the FindAllMarkers function and the default Wilcoxon rank sum test in Seurat; a minimum expression in 20% of neurons was set as the inclusion limit. One class per sheet (click through tabs at the bottom of the spreadsheet to see the different cell classes). I1 and I2 are the injured neurons; note clusters C1/2, C3, C5, C7/9/10, C8, C11, C12, C13 correspond directly to classes identified earlier (*Nguyen et al., 2017*). Clusters C4 and C6 from that analysis were divided further by the sequencing of nuclei. The new classes of these neurons have been designated C4, C6 or C4-C6 based on their UMap location and expression profiles; these are arbitrarily distinguished as (a), (b), (c)... etc. based solely on the number of neurons in a given cluster (see *Figure 5*). Note that although there are many advantages of sn-RNA sequencing, there are several drawbacks for example highly expressed ISH markers: *Mrgprd* (C13), *Mrgrpa3* (C12), *Nppb* (C11), *Trpv1* (C7/9/10 and C8), *Cd34* (C3) show far lower pct. one values (i.e. greater dropout) than in single cell analysis. Some transcripts for example *Hjurp* (C1/2) are probably not expressed genes; *Hjurp* is located adjacent to *Trpm8* (the primary marker of C1/2 cells) in the mouse

genome but was not detected at high level in sc-sequencing (*Nguyen et al., 2017*). As *Hjurp* plays a role in cell division it is unlikely to be a neural marker. There are also some minor problems associated with the alignment of reads to the pre-mRNA build of the mouse genome including clear mis-naming of a few genes (*Dlg2* is named *Dlg2.1* in this build; some transcripts map to Bacs: RP23 and RP24 etc.). The pre-mRNA basis for mapping reads also led to removal of genes that fall in intervals where genes overlap because introns are considered part of the coding sequence. One notable example of this is the pan-neuronal marker *Tubb3* that is in an interval spanned by a predicted gene (*Gm20388*) and is thus consistently missing from published mouse sn-data and our analysis. These problems are unlikely to have a strong effect on clustering or the main conclusions drawn from sn-analyses.

DOI: https://doi.org/10.7554/eLife.49679.020

• Transparent reporting form
DOI: https://doi.org/10.7554/eLife.49679.021

## Data availability

Sequencing data have been deposited in GEO under accession code GSE131272.

The following dataset was generated:

| Author(s) | Year | Dataset title | Dataset URL | Database and Identifier |
|---|---|---|---|---|
| Nguyen M, Ryba N | 2019 | Peripheral injury induces a conserved but transient transcriptional state in somatosensory neurons | https://www.ncbi.nlm.nih.gov/geo/query/acc.cgi?acc=GSE131272 | NCBI Gene Expression Omnibus, GSE131272 |

The following previously published dataset was used:

| Author(s) | Year | Dataset title | Dataset URL | Database and Identifier |
|---|---|---|---|---|
| Minh Q Nguyen, Nicholas Ryba | 2017 | Diversity amongst trigeminal neurons revealed by high throughput single cell sequencing | https://www.ncbi.nlm.nih.gov/gds/?term=GSE101984[Accession] | NCBI Gene Expression Omnibus, GSE101984 |

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
