## [Decision Letter]

Thank you for submitting your article "Stereotyped transcriptomic transformation of somatosensory neurons in response to injury" for consideration by *eLife*. Your article has been reviewed by four peer reviewers, including Peggy Mason as the Reviewing Editor and Reviewer #1, and the evaluation has been overseen by Eve Marder as the Senior Editor.

The reviewers have discussed the reviews with one another and the Reviewing Editor has drafted this decision to help you prepare a revised submission.

This work suggests that injured trigeminal neurons from animals with a wide range of injuries – from a scratch to a nerve crush – will de-differentiate into a phenotype that is common to all afferent types. The independence of phenotype from afferent type and the range of injuries that yield this are both exciting new developments. Yet the conclusions are more intriguing and appealing than the data are compelling. To do justice to this work, the authors are requested to expand, increase the quantification, and all-around sharpen the results and figures. Thus, the reviewers all believe that the authors would be best served by tightening the results up substantially. Areas that would benefit from attention include but are not limited to the following.

- Place a clear methodological description of the minimal injury that yields the common phenotypic change in the results narrative. And explicitly compare the minimal injury to sham surgery.

- A full analysis of RNAseq data, that goes beyond a list of genes in a table, such as GO, pathway analysis, transcription factor binding site analysis etc.

- Provide detailed description of the mice used and validation of the Atf3-IRES-Cre knockin mouse generated.

Please use the detailed comments of the reviewers. We look forward to a substantial revision,

Reviewer #1:

This is an exciting contribution that takes a new look at the idea that changes in gene regulation are critical to the generation of neuropathic pain. Support for this attractive idea derives from abundant studies looking at the upregulation of genes in DRG cells after peripheral injury. Here the authors look at several far more innocuous manipulations beyond the typical constriction/partial transection injury. And surprisingly these manipulations cause similar genetic changes as do the frank injuries. These findings suggest that the role of changes in gene regulation to the generation of neuropathic pain is at least more nuanced than currently formulated.

Additional findings (e.g. all types of cells adopt a pair of injured cell profiles) round out this contribution.

While these findings are exciting, this reviewer has a request and a concern:

1) The impact of these findings on the idea that changes in gene regulation are critical to the generation of neuropathic pain should be discussed explicitly. These results suggest that by not looking at more innocuous / acute manipulations, previous research may have missed the full natural history of gene expression changes after peripheral damage. Please add this to the Discussion.

2) The results are not compelling because the figures do not illustrate the results in a compelling fashion. The results wrt the sham controls are also confusing. Were there changes after sham surgeries? Why isn’t sham listed in Figure 5—figure supplement 1C.

Reviewer #2:

The authors combined single-nuclei RNAseq of isolated trigeminal neuronal nuclei with in situ hybridization (ISH) to characterize changes in gene expression after infraorbital trigeminal nerve transection (IOT) or a more moderate facial cut injury. The authors find that IOT leads to two new injury related clusters, with peak cell numbers 2-7dpi followed by a progressive decline starting 21dpi. They find that facial cut injury with or without scratching also generates two similar injury related clusters albeit with far less cells in each cluster. ISH was performed to validate these findings by showing either the presence or absence of colocalization of select genes with the injury-marker gene ATF3. Overall, the conclusions are modest, expected, and narrow in scope. Further, the RNAseq data provides little analysis beyond identification of neuronal clusters and the ISH experiments lack scientific rigor. Hence, the impact of the paper is limited. This work might be more suitable to be published as a resource rather than a research.

The major finding of the RNAseq data is that injured peripheral neurons alter their gene expression and form distinct cluster(s), whether injury occurs to the nerve or the terminations in the facial skin, and that this clustering persists yet subsides over time. This conclusion lacks novelty and rather mainly validates decades of research on this topic. No additional analysis is done with the RNAseq data (e.g., GO/Pathway Analysis).

The authors perform no quantification or statistical testing for the ISH experiments and basic controls are not used (Figure 1D-G – uninjured staining for Atf3 and select genes). In Figure 1D-G, Atf3 expression looks very high and might not match the number of injured cells reported in the sequencing (13.8%). Clarification of total number of neurons using a general neuronal marker can help.

The 'tables' provided appear to be raw data excel files generated by the sequencing software more suitable to be labeled as supplements rather than 'tables'. Tables a1 and a2 are mentioned but not included in the current version.

The sn-transcriptomics experiments represent a mix of sequencing from different platforms (DropSeq- for controls and 10X genomics for injuries). This inconsistency can lead to misinterpretation of the results. Information on sequencing depth and counts normalization is missing.

In Figure 3, there is no mention of time post injury where the ISH was done. Quantification of co-localized markers would help.

In Figure 4, does control represent sham-injury as in Figure 3?

In Figure 5, quantifications of images are necessary. Some indication on the infection rate of the virus are needed, since on day 7 it looks like only about 50% of Atf3 cells are also TdTomato positive. Details on the Atf3 -IRES-cre mice are missing, are these homozygous or heterozygous. Claiming injury does not lead to cell death of injured cells should be shown by using a dead cell marker. Using the Atf3-cre; Flx TdTomato together with markers for different neuronal cell types could shed light on the injured cells identity following an injury.

Reviewer #3:

In general this is a well written study about an interesting topic and the results are of fundamental importance. Namely that all sensory neurons collapse into close to one phenotype after injury, which is likely a transcription profile related to regeneration. Following recovery, however, they return to their original phenotypes independent of developmental cues. The conclusions of this study are important and of interest to many, those interested in nerve regeneration, sensory neuron physiology and neural development.

Within the tables please include a control and injured neuron expression levels. I.e. a p-value for inclusion in the group, a mean expression level for the cells in this group (Log2 TPM+1), a mean expression level in all other groups, and a fraction of cells in this group with non-zero expression. As in Lacar et al., 2016, and others.

Is there a reason for just showing the top 100 genes in each category in Supplementary file 3, then the top 50 in Supplementary file 2? As the full data sets will be placed on GEO please include full summary lists in the tables, to enable straightforward downstream analysis of these lists by a reader.

Please include a diagram of how the nuclear sequencing was performed early in the manuscript, as this reviewer wondered while reading this paper if there were any non-neuronal groups and if not why not? By reading the Materials and methods this became clear, but for the general understanding of the study a diagram of the experiments performed would help a great deal.

Please perform pathway analysis on the regulated gene clusters. Also on the remaining 'non-regulated' transcripts that still clustered into the I1 and I2 groups.

Would transcription factor binding site analysis demonstrate what TF binding sites are over represented in the injured groups? Can the I1 and I2 groups be separated this way?

Do the expression levels of the genes differ with different injury paradigms? Please quantify.

Are the uninjured profiles in the injured ganglia the same as their control uninjured counterparts? As there is extensive monocyte/non-neuronal activation in injured ganglia after nerve injury do the 'uninjured cell profiles' vary because of these changes and/or via other mechanisms? This study mentioned Gal but expand to the full group of genes and quantify the extent of these similarities/differences.

Related to the point above; in the long term injured cells (21 and 75 days) that have returned to 'normal' are there any retained strongly regulated gene signatures in common.

Ganglia from male and female mice were analyzed in these studies, do any sex specific transcripts separate into specific groups?

Non-neuronal cells? Do you have any data on those? The study would be significantly enhanced with the addition of such profiles.

Reviewer #4:

The manuscript by Nguyen et al. analyzes transcriptomic changes that occur at a single cell level in primary afferents upon injury. Although many groups have looked at changes in gene expression upon injury, this is the first to do so at the single cell level, and the results are extremely novel and interesting. The major conceptual advance that is revealed through these analyses is that injured neurons take on a new, somewhat de-differentiated phenotype that appears to be independent of the primary afferent cell type involved – an injured state, that is common across afferents. The study also has the potential to provide a very valuable resource to the pain community.

1) In many places, it feels like the authors are describing their impression of the data. Although I have full confidence that the authors have looked at their data carefully, the analyses would appear more rigorous if they were backed up with quantitative data, where possible.

2) In other instances, quantification is not possible. Nevertheless, a fuller description of the examples that led the authors to their interpretation would be helpful. For instance, "I1 neurons expressed markers that suggested they were originally involved in more discriminative types of sensation, while I2 neurons had an expression profile more related to nociceptors (Figure 2—figure supplement 2)" Which genes led to this interpretation?

3) Although the raw data have been made available, I think this form of data will only be useful to those who have extensive experience with clustering algorithms and wish to reanalyze all the data from scratch. It would be much better to provide the datasets in which gene expression in individual cells has been assigned to a cluster. This resource would allow people to readily look up genes of interest and mine the data for other interesting observations.

4) The mice that have scratch lesions should be described in more detail. They seem to be a double transgenic strain where TetTag, Fos-tTA drove expression of tetO-mCherry-2A-Gnaq. Were there mCherry-positive neurons in these mice that was (presumably) indicative of neurons that were driving the itch-like behavior? If so, were they primary afferents with somatotopy consistent with the lesions? Even if the authors have no idea why these mice scratch to the point of injury, I think it would be better to be more transparent about it. Also, although I acknowledge that this allele is not a major part of your study, given that you used it, I think it is important to describe it in more detail. For instance, was the mCherry-2A-Gnaq allele targeted to the Rosa locus or randomly inserted?

5) Please provide a map for the construction of the Atf3-IRES-Cre knockin mouse and some indication that you have validated it.

---

## [Author Response]

This work suggests that injured trigeminal neurons from animals with a wide range of injuries – from a scratch to a nerve crush – will de-differentiate into a phenotype that is common to all afferent types. The independence of phenotype from afferent type and the range of injuries that yield this are both exciting new developments. Yet the conclusions are more intriguing and appealing than the data are compelling. To do justice to this work, the authors are requested to expand, increase the quantification, and all-around sharpen the results and figures. Thus, the reviewers all believe that the authors would be best served by tightening the results up substantially. Areas that would benefit from attention include but are not limited to the following.- Place a clear methodological description of the minimal injury that yields the common phenotypic change in the results narrative. And explicitly compare the minimal injury to sham surgery.- A full analysis of RNAseq data, that goes beyond a list of genes in a table, such as GO, pathway analysis, transcription factor binding site analysis etc.- Provide detailed description of the mice used and validation of the Atf3-IRES-Cre knockin mouse generated.Please use the detailed comments of the reviewers. We look forward to a substantial revision,Reviewer #1:This is an exciting contribution that takes a new look at the idea that changes in gene regulation are critical to the generation of neuropathic pain. Support for this attractive idea derives from abundant studies looking at the upregulation of genes in DRG cells after peripheral injury. Here the authors look at several far more innocuous manipulations beyond the typical constriction/partial transection injury. And surprisingly these manipulations cause similar genetic changes as do the frank injuries. These findings suggest that the role of changes in gene regulation to the generation of neuropathic pain is at least more nuanced than currently formulated.Additional findings (e.g. all types of cells adopt a pair of injured cell profiles) round out this contribution.While these findings are exciting, this reviewer has a request and a concern:1) The impact of these findings on the idea that changes in gene regulation are critical to the generation of neuropathic pain should be discussed explicitly. These results suggest that by not looking at more innocuous / acute manipulations, previous research may have missed the full natural history of gene expression changes after peripheral damage. Please add this to the Discussion.2) The results are not compelling because the figures do not illustrate the results in a compelling fashion. The results wrt the sham controls are also confusing. Were there changes after sham surgeries? Why isn’t sham listed in Figure 5—figure supplement 1C.Reviewer #2:The authors combined single-nuclei RNAseq of isolated trigeminal neuronal nuclei with in situ hybridization (ISH) to characterize changes in gene expression after infraorbital trigeminal nerve transection (IOT) or a more moderate facial cut injury. The authors find that IOT leads to two new injury related clusters, with peak cell numbers 2-7dpi followed by a progressive decline starting 21dpi. They find that facial cut injury with or without scratching also generates two similar injury related clusters albeit with far less cells in each cluster. ISH was performed to validate these findings by showing either the presence or absence of colocalization of select genes with the injury-marker gene ATF3. Overall, the conclusions are modest, expected, and narrow in scope. Further, the RNAseq data provides little analysis beyond identification of neuronal clusters and the ISH experiments lack scientific rigor. Hence, the impact of the paper is limited. This work might be more suitable to be published as a resource rather than a research.The major finding of the RNAseq data is that injured peripheral neurons alter their gene expression and form distinct cluster(s), whether injury occurs to the nerve or the terminations in the facial skin, and that this clustering persists yet subsides over time. This conclusion lacks novelty and rather mainly validates decades of research on this topic. No additional analysis is done with the RNAseq data (e.g., GO/Pathway Analysis).The authors perform no quantification or statistical testing for the ISH experiments and basic controls are not used (Figure 1D-G – uninjured staining for Atf3 and select genes). In Figure 1D-G, Atf3 expression looks very high and might not match the number of injured cells reported in the sequencing (13.8%). Clarification of total number of neurons using a general neuronal marker can help.The 'tables' provided appear to be raw data excel files generated by the sequencing software more suitable to be labeled as supplements rather than 'tables'. Tables a1 and a2 are mentioned but not included in the current version.The sn-transcriptomics experiments represent a mix of sequencing from different platforms (DropSeq- for controls and 10X genomics for injuries). This inconsistency can lead to misinterpretation of the results. Information on sequencing depth and counts normalization is missing.In Figure 3, there is no mention of time post injury where the ISH was done. Quantification of co-localized markers would help.In Figure 4, does control represent sham-injury as in Figure 3?In Figure 5, quantifications of images are necessary. Some indication on the infection rate of the virus are needed, since on day 7 it looks like only about 50% of Atf3 cells are also TdTomato positive. Details on the Atf3 -IRES-cre mice are missing, are these homozygous or heterozygous. Claiming injury does not lead to cell death of injured cells should be shown by using a dead cell marker. Using the Atf3-cre; Flx TdTomato together with markers for different neuronal cell types could shed light on the injured cells identity following an injury.Reviewer #3:In general this is a well written study about an interesting topic and the results are of fundamental importance. Namely that all sensory neurons collapse into close to one phenotype after injury, which is likely a transcription profile related to regeneration. Following recovery, however, they return to their original phenotypes independent of developmental cues. The conclusions of this study are important and of interest to many, those interested in nerve regeneration, sensory neuron physiology and neural development.Within the tables please include a control and injured neuron expression levels. I.e. a p-value for inclusion in the group, a mean expression level for the cells in this group (Log2 TPM+1), a mean expression level in all other groups, and a fraction of cells in this group with non-zero expression. As in Lacar et al., 2016, and others.Is there a reason for just showing the top 100 genes in each category in Supplementary file 3, then the top 50 in Supplementary file 2? As the full data sets will be placed on GEO please include full summary lists in the tables, to enable straightforward downstream analysis of these lists by a reader.Please include a diagram of how the nuclear sequencing was performed early in the manuscript, as this reviewer wondered while reading this paper if there were any non-neuronal groups and if not why not By reading the Materials and methods this became clear, but for the general understanding of the study a diagram of the experiments performed would help a great deal.Please perform pathway analysis on the regulated gene clusters. Also on the remaining 'non-regulated' transcripts that still clustered into the I1 and I2 groups.Would transcription factor binding site analysis demonstrate what TF binding sites are over represented in the injured groups? Can the I1 and I2 groups be separated this way?Do the expression levels of the genes differ with different injury paradigms? Please quantify.Are the uninjured profiles in the injured ganglia the same as their control uninjured counterparts? As there is extensive monocyte/non-neuronal activation in injured ganglia after nerve injury do the 'uninjured cell profiles' vary because of these changes and/or via other mechanisms? This study mentioned Gal but expand to the full group of genes and quantify the extent of these similarities/differences.Related to the point above; in the long term injured cells (21 and 75 days) that have returned to 'normal' are there any retained strongly regulated gene signatures in common.Ganglia from male and female mice were analyzed in these studies, do any sex specific transcripts separate into specific groups?Non-neuronal cells? Do you have any data on those? The study would be significantly enhanced with the addition of such profiles.Reviewer #4:The manuscript by Nguyen et al. analyzes transcriptomic changes that occur at a single cell level in primary afferents upon injury. Although many groups have looked at changes in gene expression upon injury, this is the first to do so at the single cell level, and the results are extremely novel and interesting. The major conceptual advance that is revealed through these analyses is that injured neurons take on a new, somewhat de-differentiated phenotype that appears to be independent of the primary afferent cell type involved – an injured state, that is common across afferents. The study also has the potential to provide a very valuable resource to the pain community.1) In many places, it feels like the authors are describing their impression of the data. Although I have full confidence that the authors have looked at their data carefully, the analyses would appear more rigorous if they were backed up with quantitative data, where possible.2) In other instances, quantification is not possible. Nevertheless, a fuller description of the examples that led the authors to their interpretation would be helpful. For instance, "I1 neurons expressed markers that suggested they were originally involved in more discriminative types of sensation, while I2 neurons had an expression profile more related to nociceptors (Figure 2—figure supplement 2)" Which genes led to this interpretation?3) Although the raw data have been made available, I think this form of data will only be useful to those who have extensive experience with clustering algorithms and wish to reanalyze all the data from scratch. It would be much better to provide the datasets in which gene expression in individual cells has been assigned to a cluster. This resource would allow people to readily look up genes of interest and mine the data for other interesting observations.4) The mice that have scratch lesions should be described in more detail. They seem to be a double transgenic strain where TetTag, Fos-tTA drove expression of tetO-mCherry-2A-Gnaq. Were there mCherry-positive neurons in these mice that was (presumably) indicative of neurons that were driving the itch-like behavior? If so, were they primary afferents with somatotopy consistent with the lesions? Even if the authors have no idea why these mice scratch to the point of injury, I think it would be better to be more transparent about it. Also, although I acknowledge that this allele is not a major part of your study, given that you used it, I think it is important to describe it in more detail. For instance, was the mCherry-2A-Gnaq allele targeted to the Rosa locus or randomly inserted?5) Please provide a map for the construction of the Atf3-IRES-Cre knockin mouse and some indication that you have validated it.

The editorial summary requested additional quantification and general sharpening of the results to better highlight the findings and do justice to the work. We have added significant new data and analysis and have edited the text taking all the individual reviewers’ comments into account as follows:

1) Reviewer 1 requested Figure 1—figure supplement 1 be added to the main figures and reviewer 3 wanted a diagram of how the nuclear sequencing was performed: these are now included in a new Figure 1. To accommodate this change we added a short section explaining the uninjured control results and highlighting advantages of nuclear sequencing vs. cell sequencing (see also new Figure 1—figure supplement 1) for studying injury related changes in trigeminal neurons.

2) We have now provided quantitation, and statistical analysis of ISH data as requested by all reviewers. This is detailed in Supplementary file 1; the quantitation fully supports conclusions drawn from ISH results in the manuscript.

3) We have added uninjured controls to Figure 2 (old Figure 1) and Figure 4 (old Figure 3) as requested by reviewers 1 and 2. We also relabeled Figure 4 and hope that the new labeling making clear that sham IOT surgery is a mild peripheral injury rather than a control clears up any confusion about these data. That sham surgery is a mild injury model was stated in the original text, but we understand that readers often concentrate on the figures. We have also made sure colors are explained and time after injury is detailed wherever these were missing.

4) We have added GO-pathway analysis as requested by the editor and reviewers 2 and 3: Figure 2—figure supplement 3 and Figure 5—figure supplement 1. These support the conclusions about differences in the I1 and I2 classes of injured neurons as well as the role of upregulated vs. downregulated genes. The new Supplementary file 2 goes with Figure 2—figure supplement 3 and provides additional analysis of I1 vs. I2 requested by reviewer 3. Unfortunately, transcription factor analysis that this reviewer suggested might be interesting was rather uninformative (with many rather general and overlapping binding sites) and was thus not included.

5) We have added Figure 6—figure supplement 1 detailing the generation and validation of Atf3-IRES-Cre knockin mice as requested by the editor and reviewers 2 and 4 as well as more detail about the mice with scratch lesions (in the Materials and methods section).

6) We have added Figure 2—figure supplement 2 which augments quantitation of gene expression changes in the injury models as requested by reviewer 3. We have also explained in the text that the clustering and UMAP representation of the data provide analysis of gene expression based on the quantitative expression of many genes, thus the co-clustering of the different injury models is very significant. The clustering approach we used (Butler et al., 2018) minimizes impact of technical variation without forcing cells into shared clusters assuaging concerns of reviewer 2.

7) We added a clear description of the mild injuries that induce the I1/I2 injured state in the results and add comparison to the sham IOT surgery (another mild injury) as requested by the editor.

8) Reviewers 3 and 4 wanted more extensive data for readers: as requested we have included the full set of genes in Supplementary files 2-4, and as reviewer 3 wanted these include p-value for inclusion, adjusted p-value, relative expression in a class vs. other classes, fractional expression in the class in question, fractional expression in other classes.

9) As requested by the editor we added discussion focused on the idea that changes in gene regulation are critical to the generation of neuropathic pain and highlight that by not looking at more innocuous / acute manipulations, previous research may have missed the full natural history of gene expression changes after peripheral damage.

10) We supply higher resolution figures (note that there are size limits for the first review that degraded quality).